# Climatological fields of Southern Ocean interior carbonate system parameters and anthropogenic CO₂ reconstructed and integrated from float- and ship-based observations

Wanqin Zhong[1,2,3]†, Ma Xin[3,4]†, Yingxu Wu[1,2]*, Chenglong Li[1], Tianqi Shi[5], Wei Gong[4,5], Di Qi[1,2]*

[1]Polar and Marine Research Institute, Jimei University, Xiamen 361021, China
[2]Southern Marine Science and Engineering Guangdong Laboratory (Zhuhai), Zhuhai 519082, China
[3]State Key Laboratory of Information Engineering in Surveying, Mapping, and Remote Sensing, Wuhan University, Wuhan 430079, China
[4]Wuhan Institute of Quantum Technology, Wuhan 430079, China
[5]Laboratoire des Sciences du climat et del'Environnement, LSCE/IPSL, France
[6]Luojia Laboratory, Wuhan University, Wuhan 430079, China

*Wanqin Zhong and Xin Ma contributed equally to this work.*

*Correspondence to*: Yingxu Wu (yingxu.wu@jmu.edu.cn), Di Qi (qidi@jmu.edu.cn)

**Abstract.** The Southern Ocean plays a crucial role in regulating atmospheric carbon dioxide ($CO_2$) concentrations and modulating the global oceanic carbon cycle, thereby substantially mitigating the effects of anthropogenic climate change. However, due to the region's challenging environment and sparse observational coverage, large uncertainties remain regarding the magnitude and mechanisms of carbon uptake in the Southern Ocean. In recent decades, the deployment of Argo float arrays has facilitated autonomous and continuous profiling of hydrographic and biogeochemical properties from the surface to depths of up to 6,000 m, complementing traditional ship-based observations. Nevertheless, high-resolution, integrated datasets that combine ship-based and Argo-derived observations remain rare, partly due to the challenges of data harmonization, quality control, and uncertainty estimation, as well as the indirect nature of carbonate system parameter retrievals from Argo measurements. Here, we present a comprehensive, quality-controlled reconstruction of key carbonate system parameters in the Southern Ocean interior—including total alkalinity (TA), dissolved inorganic carbon (DIC), pH (total scale), nitrate ($NO_3$), phosphate ($PO_4$), silicate ($SiO_4$), anthropogenic carbon ($C_{ant}$), and aragonite saturation ($\Omega_{ar}$)—by leveraging machine learning techniques and integrating all available Argo float profiles with ship-based survey data. The resulting datasets are gridded at 1°×1° horizontal resolution and 84 vertical pressure levels (0-5,600 m), and are provided as distinct climatological products: the Float Grid (using all Argo float profiles) and the All-Data Grid (integrating all available Argo and ship-based observations). The Float Grid is further separated into the Non-$O_2$-Float Grid (limited to Core Argo floats) and $O_2$-Float Grid (limited to oxygen-measured Biogeochemical Argo floats). Each gridded product is accompanied by uncertainty estimates. The climatological products cover nearly the whole Sothern Ocean based on direct measurements instead of applying interpolating

mapping methods, thereby providing a more robust result. Model performance is assessed through cross-comparison of Argo and shipboard measurements. The gridded products, collectively termed SOCOML (Southern Ocean $CO_2$ Machine Learning products), are freely available for downloaded (doi: 10.17632/xzr59ngmpz.2) and are expected to support future studies of Southern Ocean carbon cycle.

## 1 Introduction

The Southern Ocean (south of 30°S) plays a pivotal role in the global carbon cycle by facilitating anthropogenic carbon uptake from the atmosphere and transporting to the ocean interior (Morrison et al., 2022), thereby modulating $CO_2$ concentrations from past climates to the present and into the future (Hauck et al., 2023). Since industrialization, rising atmospheric $CO_2$ concentration has been the primarily driver of the strengthening ocean carbon sink, with the Southern Ocean accounting for around one-quarter of the anthropogenic carbon ($C_{ant}$) uptake (Gruber et al., 2019a). Oceanic carbon uptake is fundamentally constrained by the amount of carbon in the upper ocean and by the rate at which $C_{ant}$, in the form of dissolved inorganic carbon (DIC), is transported into the ocean interior (Bopp et al., 2015). The large-scale upwelling limb of the meridional overturning circulation (MOC) in the mid-latitude Southern Ocean enables the uptake of excess carbon and its subsequent transport northward into the upper ocean (Marshall & Speer, 2012; Pellichero et al., 2018) or southward to fill the global abyssal carbon reservoir (Rios et al., 2012; Pardo et al., 2017; Mahieu et al., 2020; Zhang et al., 2023). Both carbon transport pathways in mid-latitude and high-latitude Southern Ocean are interconnected via the global thermohaline circulation, contributing to the removal of anthropogenic carbon from the surface ocean.

The continuous uptake of $C_{ant}$ by the ocean leads to declines in seawater pH and calcium carbonate ($CaCO_3$) saturation, collectively referred to as ocean acidification (OA) (Doney et al., 2009). In the Southern Ocean, substantial $CO_2$ uptake causes buffering capacity and aragonite saturation states ($\Omega_{ar}$) to decline faster than the global average (Orr et al., 2005; Petrou et al., 2019). Recent multidecadal studies found reinvigoration of carbon sink since 2000s (Landschützer et al., 2015; Zemskova et al., 2022) and pronounced acidification particularly in the Antarctic Zone (Bednaršek et al., 2012; Xue et al., 2018). To quantitatively assess and understand the underlying feedback mechanism involved in carbon uptake and storage, sustained high-quality oceanic measurements across timescales and the entire Southern Ocean are highly needed. Key oceanic interior variables of the carbonate system—total alkalinity (TA), dissolved inorganic carbon (DIC), and pH—each has strengths for explaining climate change process. For example, increasing DIC from $C_{ant}$ storage leads to pH reduction, while TA reflects the ocean's capacity to buffer pH changes (Orr et al., 2005). Moreover, measuring nutrient concentration (nitrate, phosphate and silicate) is also associated to the oceanic biogeochemical process (e.g., involved in the calculation of seawater carbonate chemistry and $C_{ant}$ (Gruber et al., 1996; van Heuven et al., 2011)). Therefore, a comprehensive dataset that combines TA, DIC, pH, and nutrients offers detailed insights into the variability of ocean carbon sink (characterized by $C_{ant}$), the progression of OA (characterized by $\Omega_{ar}$), and its potential impacts on marine ecosystems (Doney et al., 2020; Gruber et al., 2019b; Kroeker et al., 2013; Sabine et al., 2004).

Despite its importance in the global carbon cycle, the vast and remote nature of the Southern Ocean severely limits observational coverage, especially with regard to biogeochemical variables. Two major databases compile shipboard measurements: the Surface Ocean $CO_2$ Atlas (SOCAT, Bakker et al., 2016) provides a quality-controlled dataset of the $CO_2$ fugacity for the global surface ocean and coastal seas, while the Global Ocean Data Analysis Project version 2 (GLODAPv2, Olsen et al., 2016) offers quality-controlled data as well as climatological products (Lauvset et al, 2016) from the surface into the ocean interior, including TA and DIC. However, the scarcity of shipboard measurements, particularly during austral winter, leads to large uncertainty in evaluating the Southern Ocean carbon sink (Friedlingstein et al., 2025; Hauck et al., 2023; Lo Monaco et al., 2005). Measurements of more difficult-to-observe variables, such as TA, DIC and pH, are particularly scarce, comprising only about half of the data available for other variables in GLODAPv2 database (Figure 1d).

Novel observations recently collected by profiling floats, as part of the Argo program, have revolutionized the ability to monitor the Southern Ocean since the 2000s (Riser et al., 2016; Silvano et al., 2023). These autonomous floats, including Core-Argo and Biogeochemical Argo (BGC-Argo), measure seawater properties (temperature, salinity, and pressure) and optional biogeochemical variables (oxygen, pH, and nitrate, normally for BGC-Argo) between the surface and depths of 2,000 m, with Deep-Argo floats reaching depths of up to 6,000 m. The rapid increase in BGC-Argo floats has significantly expanded the amount of carbonate system data and thus revealed spatial and temporal variability with depth globally or regionally (Williams et al., 2017; Wu et al., 2022; Wu & Qi, 2023). Despite transformative potential, BGC-Argo floats currently constitute less than a quarter of all Argo floats in the Southern Ocean (Figure 1). This limited coverage highlights the need to develop robust methods for deriving carbonate system parameters from all Argo observations, which would greatly improve and supplement current observation-based $CO_2$ datasets and support more comprehensive monitoring of ocean carbon dynamics.

Multiple efforts have focused on retrieving carbonate chemistry variables by utilizing the strong regional correlations among seawater properties and by estimating carbonate chemistry variables using combinations of more readily available variables, such as temperature, salinity, and dissolved oxygen. This approach is effective because oceanographic processes influence the distributions of many seawater properties in similar ways, allowing algorithms to be trained to reproduce carbonate system parameters from co-located measurements of other seawater properties (Carter et al., 2021a). Among the primary methods, multilinear regression (MLR) and neural networks (NN) are widely used to estimate various seawater properties, including nutrients and carbonate chemistry variables. MLR models, such as LIAR (Locally Interpolated Alkalinity Regression, (Carter et al., 2021b; Carter et al., 2016)), are straightforward and interpretable but are limited to capturing linear relationships. In contrast, neural network approaches, like Bayesian neural network (BNN)-based method (CANYON-B, Bittig et al., 2018; Sauzède et al., 2017) can model more complex, nonlinear patterns and often provide higher accuracy. Building on MLR and NN methods, the ESPER_LIR and ESPER_NN routines were recently introduced to further expand predictive capabilities. For instance, Asselot et al., 2024 applied the ESPER_NN method to reconstruct $C_{ant}$ from Argo data, demonstrating the combination of Argo float observations with machine-learning approaches offers new perspectives and robust insights into the storage and transport of $C_{ant}$ in the interior ocean.

In this study, we leverage ESPER_NN model, integrating the high-accuracy GLODAPv2 database with profiling measurements of fine spatiotemporal resolution, to generate a comprehensive carbonate system dataset throughout the interior Southern Ocean that extends from the surface to the deep ocean (5,600 m). TA, DIC, pH (total scale), nitrate ($NO_3$), phosphate ($PO_4$), silicate ($SiO_4$), and $O_2$ (when direct measurements are unavailable) are obtained through neural networks, while $\Omega_{ar}$ is computed using CO2SYS based on reconstructed TA and DIC. And $C_{ant}$ is estimated using TrOCA method (Zhang et al., 2023 and Metzl et al., 2024 doing the same with TrOCA in Southern Ocean). We refer to the data products as Southern Ocean $CO_2$ Machine Learning products (SOCOML, doi: 10.17632/xzr59ngmpz.2). The rest of the paper describes the data and methodology used in the estimation of dataset for ocean carbon research. This is followed by the assessment and climatological variability of the dataset. Last, we discuss the uncertainty estimation process and potential influence.

## 2 Data used and reprocessing

### 2.1 GLODAP

The Global Ocean Data Analysis Project (version GLODAPv2.2023), a bias-corrected observational ocean biogeochemical dataset, serves as the ship-based observational data source for this study. Data from GLODAPv2.2023 collected south of 30°S are selected, including concurrent measurements of hydrographic properties, nutrients ($NO_3$, $PO_4$ and $SiO_4$), and carbonate system parameters (TA, DIC, and pH), as detailed in Table 1. Before reprocessing the data, two cruises (Expocode: 316N19871123 and 318M19771204) were excluded due to noisy at depth or large quality control (QC) adjustments, as reported by (Carter et al., 2021a). Subsequently, the remaining 160 cruises undergo secondary QC and adjustment check. Measurements flagged as poor quality includes TA and DIC values with adjustments exceeding $\pm$ 10 $\mu$mol kg$^{-1}$, pH adjustments greater than $\pm$ 0.015 pH units, and nutrient data with multiplicative adjustments surpassing 10% (Carter et al., 2018; Olsen et al., 2016). The precise adjustment values are documented in the GLODAPv2 Adjustment Table, accessible at https://glodapv2.geomar.de/. Following this QC step, five cruises are excluded for TA, three cruises (1,112 measurements) for DIC, one cruise (1,474 measurements) for $PO_4$, and one cruise (940 measurements) for $SiO_4$ (see detailed exclusions in Table A1). Although no significant offsets were identified in pH measurements, one cruise (Expocode: 49HG19950414) are excluded as noted by (Carter et al., 2018). Importantly, quality control is performed independently for each variable. Subsequently, TA and DIC measurements are retained only when nutrient observations are available, following Carter et al. (2021a). The pH data in GLODAPv2 comprise a mixture of spectrophotometric- and potentiometric-derived measurements. To ensure data consistency, pH measurements are homogenized to align with pH calculated from TA and DIC, following (Carter et al., 2018). Classification of pH data are conducted based on documentation available from https://cchdo.ucsd.edu/, as shown in Table A2 and Figure A1.

The ESPER_LIR and ESPER_NN model were trained using data from GLODAPv2.2020, whereas the CANYON-B model utilized the original GLODAPv2 release. Assessment dataset for model performance comparison is identified from the

GLODAPv2.2023, consisting of cruises added subsequent to the GLODAPv2.2020 release (i.e., cruise numbers ≥2107). Initial

130 comparative analysis among CANYON-B, ESPER_NN, and ESPER_LIR models is conducted using this assessment data.

**Table 1.** Numbers of shipboard GLODAPv2 measurements and Argo float profiles for each variable in the Southern Ocean used in this study. The assessment dataset of GLODAPv2 data product used for mode-performance comparisons contains cruises added after the GLODAPv2.2020 release—specifically, those with cruise identifiers ≥ 2107.

|  | Variables | Oxygen (Y/N) | Assessment dataset | Total |
|---|---|---|---|---|
| Shipboard GLODAPv2 measurements | TA | N | 15,059 | 103,140 |
|  |  | Y | 14,954 | 101,870 |
|  | DIC | N | 15,376 | 129,799 |
|  |  | Y | 15,270 | 126,312 |
|  | pH | N | 14,996 | 56,597 |
|  |  | Y | 14,894 | 56,411 |
|  | $NO_3$ | N | 18,796 | 232,771 |
|  |  | Y | 18,575 | 226,665 |
|  | $PO_4$ | N | 18,352 | 224,262 |
|  |  | Y | 18,086 | 218,876 |
|  | $SiO_4$ | N | 19,011 | 242,736 |
|  |  | Y | 18,745 | 234,807 |
| Argo float profiles | Temp, Sal | - | - | 647,650 |
|  | Temp, Sal, Oxy | - | - | 73,296 |

## 2.2 Argo data preparation and description

135 The Argo float data were download from the Argo Data Assembly Canters (GDACs; FTP: /ifremer/argo/dac/; latest access on 23 Feb 2025) and processed using adapted code from the SAGEO2 toolbox. This dataset comprises three types of Argo floats (Core Argo, BGC-Argo, and Deep Argo) for reconstructing carbonate system parameters and nutrients using models. Since 2000, the Core Argo network has provided high-resolution temperature and salinity profiles with broad coverage (0-2,000 dbar at 10-day intervals), forming the foundation for extensive studies of oceanographic processes. Building upon this framework,

140 the BGC-Argo extends observational capabilities by employing biogeochemical sensors to measure oxygen, pH, and nitrate. To address ongoing uncertainties regarding deep ocean, the recent deployment of Deep Argo floats enables data collection down to 6,000 dbar in targeted Southern Ocean basins, providing unprecedented insights into carbon dynamics in abyssal waters.

Rigorous quality control leads to the exclusion of three categories of problematic data: 1) floats on the Argo Program's grey

145 list identified for sensor drift or transmission errors; 2) floats with 10 or fewer operational cycles, due to insufficient calibration

stability; and 3) aberrant profiles with incomplete measurements. Additionally, only adjusted data flagged as "Good" or "Probably Good" (QC flags 1 and 2, respectively) were included. The remaining floats were systematically classified based on the presence or absence of oxygen data. This classification yielded two distinct float categories, underpinning our dual-pathway analytical approach and ensuring robust estimation across diverse observational regimes. Overall, this study includes data from 4,346 Argo floats, of which 525 are equipped with oxygen sensor providing 73,296 profiles, and the remaining 3,821 floats without oxygen sensor providing 647,650 profiles (Table 1).

There are substantial spatial sampling gaps in the high-quality GLODAP data, particularly in the high-latitude Southern Ocean (Figure 1). Furthermore, Figure 1 reveals a pronounced seasonal bias toward the austral summer, with nearly four times as many measurements collected during this period compared to winter. In contrast, Argo floats provide extensive spatiotemporal coverage, owing to their flexible deployment and consistent ten-day sampling cycles. Although the number of Argo floats equipped with oxygen sensors have increased greatly in recent decades (Figure 1f), the Argo observational network is still predominantly composed of Core-Argo floats without oxygen sensors, which constitute over 85% of the dataset and achieve nearly complete spatial coverage across the Southern Ocean (Figure 1a, c). The broad coverage offers an unprecedented foundation for reconstructing carbon system dynamics in the region. However, because of current limitations in data quality and correction methods (Maurer et al., 2021; Williams et al., 2017), nitrate and pH measurements from BGC-Argo floats are not used in this study; only temperature, salinity, and $O_2$ are employed. Ongoing improvements in quality control and correction procedures may enable the incorporation of these measurements in future studies.

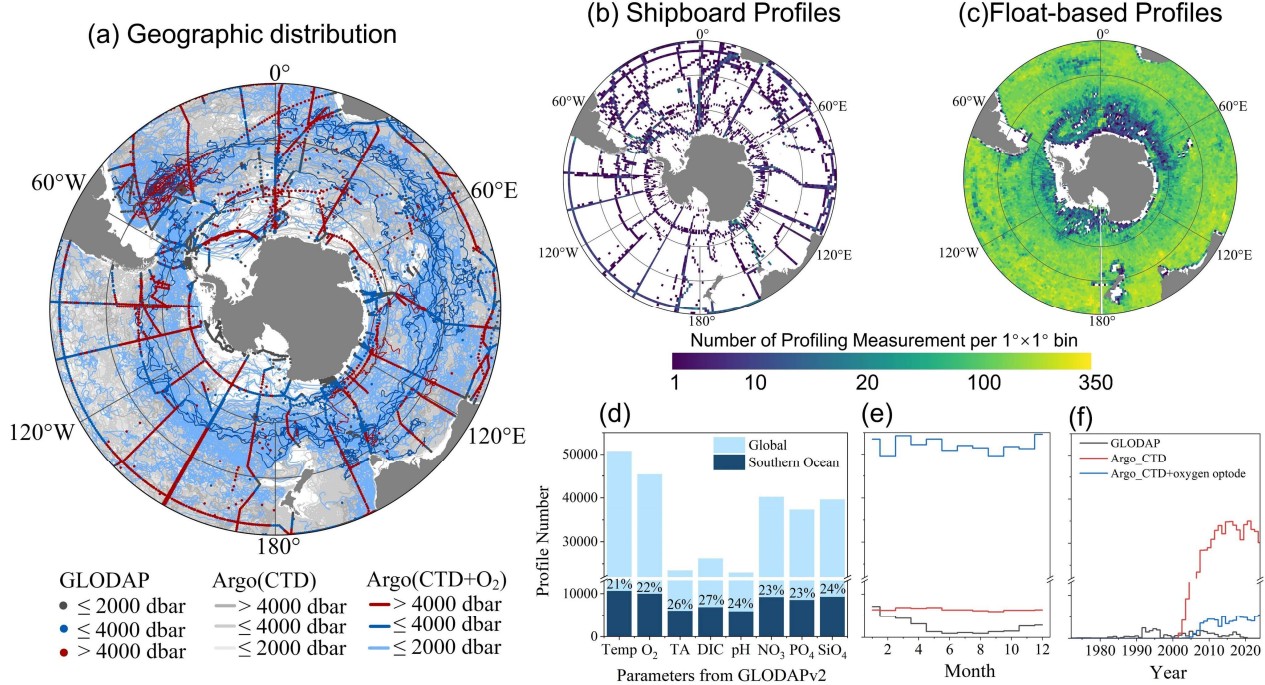

**Figure 1.** Spatial and temporal coverage of the measurements in 7,607 stations (dots) from the GLODAPv2.2023 database (Lauvset et al., 2024) and 4,603 Argo profiling floats (lines). (a): Geographic distribution of GLODAP, Argo (CTD), and Argo (CTD+$O_2$) which is categorized into three types according to the maximum pressure of observation. (b-c): Number of profiling measurement covered by shipboard (GLODAP) and float-based (Argo) observations for the entire period since 1972 per 1° × 1° bin. (c): The parameter types, seasonal, and latitudinal distribution of profiles from all three dataset (grey for GLODAP, blue for Argo only with CTD, and red for Argo with CTD and oxygen sensor). The number of GLODAP profiles in (e-f) has been multiplied by a factor of 3 for visibility.

## 3 Methodology

### 3.1 Reconstruction of carbonate system parameters and nutrients

This study employs a dynamically adaptive framework to reconstruct carbonate system parameters and nutrients by integrating heterogeneous Argo float observations with high-quality GLODAP measurements. Figure 2 illustrates the overall workflow for generating gridded products in the Southern Ocean. Based on performance comparisons (see Section 4.1 for detail), the best-performing model is applied to reconstruct key biogeochemical tracers (oxygen, nitrate, phosphate, silicate) as well as carbonate system parameters (TA, DIC, pH). To accommodate differences in observational capabilities among Argo floats, particularly regarding the presence or absence of oxygen sensors, input observations are dynamically sorted into two reconstruction pathways:

1. *Full-parameter* pathway (green in Figure 2): This pathway utilizes all available measured variables, including hydrographic properties, and dissolved oxygen concentrations from floats equipped with oxygen sensors.

2. *Hydrography-only* pathway (blue in Figure 2): This pathway reconstructs targeted variables and oxygen concentration based solely on CTD measurements (salinity, temperature, depth) from floats lacking oxygen sensors.

The resulting dataset includes both reconstructed variables (derived indirectly from Argo profiles) and direct high-quality ship-based observations.

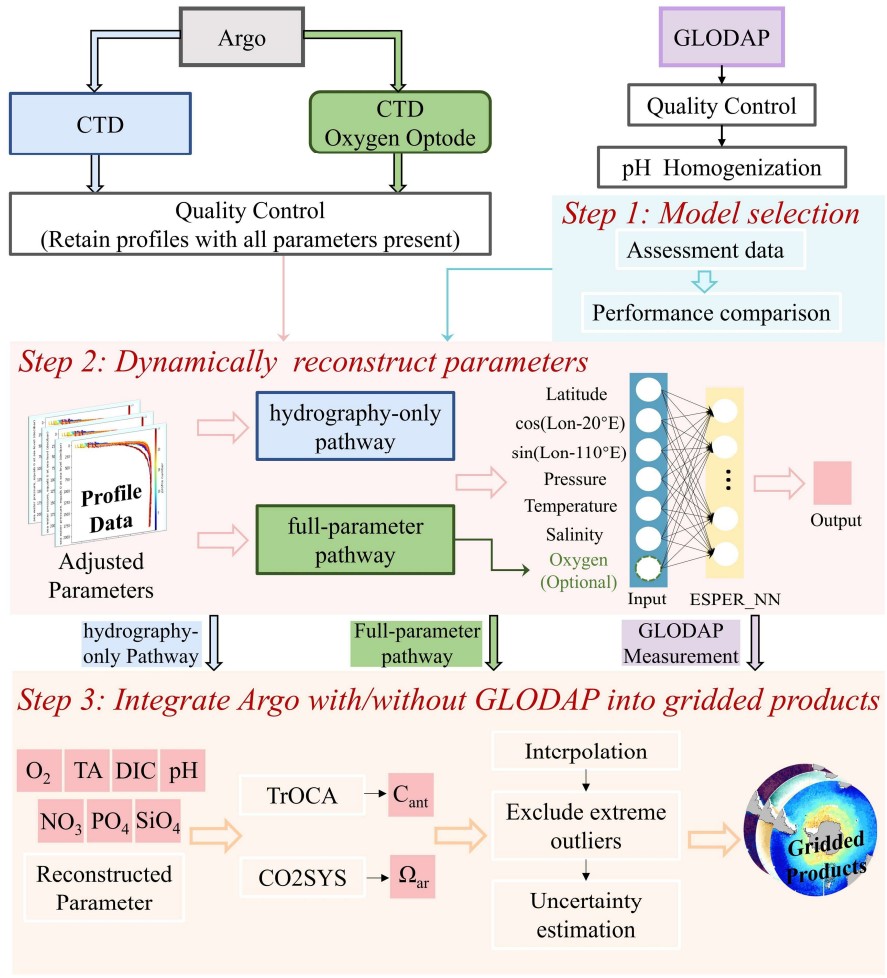

**Figure 2.** Overall workflow for generating interior carbonate system gridded products in Southern Ocean. The top panel shows data inputs: Argo with CTD only (in blue), Argo with CTD and oxygen sensor (in green), and GLODAP (in purple). All data undergo quality control procedures. The workflow comprises three main steps: the top panel depicts model selection (CANYON-B, ESPER_LIR, and ESPER_NN); the middle panel illustrates the use of ESPER_NN to predict carbonate system parameters with or without oxygen data; the bottom panel shows the integration of Argo/GLODAP data into gridded products with derived anthropogenic $CO_2$ and aragonite saturation data (see Section 3.2 for detailed calculations).

### 3.2 Estimation of anthropogenic carbon ($C_{ant}$) and aragonite saturation state ($\Omega_{ar}$)

Typical methods for calculating $C_{ant}$ from DIC measurements include the $\Delta C^*$ method (Gruber et al., 1996; Gruber et al., 1998; Sabine et al., 2004), the extended multiple linear regression (eMLR) method (Gruber et al., 2019a), and the Tracer combining Oxygen, inorganic Carbon, and total Alkalinity (TrOCA) method (F. Touratier et al., 2007; Franck Touratier & Goyet, 2004) has been widely applied. Although the $\Delta C^*$ method has been widely used to quantify $C_{ant}$ inventory, it relies on parameter

customization through Optimum Multiparameter (OMP) analysis, which requires direct nutrient measurements of $NO_3$, $PO_4$, and $SiO_4$—data not available in our dataset. The eMLR method relies on repeat hydrographic measurements (Friis et al., 2005), which are not available for Argo profiles, and its output reflects temporal change in $C_{ant}$ rather than absolute concentration. For these reasons, this study employs the TrOCA method, which is relatively straightforward, extensively utilized in Southern Ocean studies (Metzl et al., 2024; Zhang et al., 2023), and has been demonstrated to be reliable through comparative analyses (Lo Monaco et al., 2005; Vázquez-Rodríguez et al., 2009; Mahieu et al., 2020; Zhang et al., 2023).

$$TrOCA = O_2 + a(DIC - \frac{1}{2}TA) \tag{1}$$

$$C_{ant} = \frac{TrOCA - TrOCA^0}{a} = \frac{O_2 + 1.279(DIC - \frac{1}{2}TA) - e^{7.511 - (1.087 \times 10^{-2})\theta - \frac{7.81 \times 10^5}{TA^2}}}{1.279} \tag{2}$$

where $\theta$ is potential temperature, °C; $O_2$ is dissolved oxygen, $\mu mol\ kg^{-1}$; DIC is total inorganic carbon, $\mu mol\ kg^{-1}$; TA is total alkalinity, $\mu mol\ kg^{-1}$. In the *full-parameter* pathway, calculated values utilized observed oxygen concentrations alongside model-derived TA and DIC estimates. Conversely, the *hydrography-only* pathway employed model-derived estimates for all three variables (oxygen, TA, and DIC). Finally, the $C_{ant}$ values are scaled to the reference year 2013 to deal with exponential increase of anthropogenic $CO_2$ burden in the climatological products (see Section 3.3) (Carter et al., 2021a; Tanhua et al., 2007). A detailed description of the scaling method is given in the Appendix B1. It should be noted that the TrOCA approach is limited to waters below the euphotic layer (Lo Monaco et al., 2005), therefore the $C_{ant}$ estimates above 100 m are excluded from this dataset.

The aragonite saturation state ($\Omega_{ar}$) is calculated using the CO2SYS software (v3, Sharp et al., 2023), requiring TA, DIC, temperature, salinity, and pressure as inputs to minimize uncertainty in the results (Orr et al., 2018). The following thermodynamic parameterizations are employed: carbonic acid dissociation constants from Lueker et al., 2000, hydrogen fluoride (HF) dissociation constants from Perez & Fraga, 1987, the ratio of total boron ($B_T$) to practical salinity ($S_p$) from (Lee et al., 2010), and bisulfate dissociation constants ($KHSO_4$) from Dickson et al., 1990.

## 3.3 Construction of gridded products

Profile data for each parameter are sorted into spatial bins of 1° longitude × 1° latitude bins and 84 vertical levels to generate homogenized three-dimensional gridded products. Data derived from both float- and ship-based observations are integrated into this spatial framework, ensuring robust spatial and depth coverage. To maximize data density, we construct an "All-Data Grid" by merging all available reconstructions and observations. In addition, three specialized gridded products are generated: the "Float Grid", comprising only float-based reconstructions; the "Non-$O_2$-Float Grid", limited to floats without oxygen measurements; and the "$O_2$-Float Grid", limited to BGC-Argo floats. The latter two grids facilitate sensitivity analyses of oxygen's influence on carbonate system parameter reconstructions. All these gridded datasets serve as the basis for subsequent analyses.

Figure 3 demonstrates the vertical sampling spacing of CTD and dissolved oxygen from Argo floats. Typically, floats sample at intervals of 10 m or finer from the surface down to 200 m and at intervals of 50 m or finer between 500 to 2000 m. Floats equipped with oxygen sensors sample dissolved oxygen at higher resolution. Measured CTD profiles are prioritized, but interpolated profiles are used when concurrent oxygen data are unavailable. To align with the float sampling scheme and maximize data utilization, the water column (0-5,600 m) is divided into 84 vertical depth levels (highlighted in yellow in Figure 3): 0–100 m at 5 m intervals (20 levels), 100–500 m at 25 m intervals (15 levels), 500–2000 m at 50 m intervals (30 levels), and 2000–5600 m at 200 m intervals (19 levels). The deepest level (5600 m) corresponds to the maximum float measurement depth.

Each Argo float profile is interpolated to these predefined vertical levels using the Piecewise Cubic Hermite Interpolating Polynomial routine ("intprofile.m" in the 2nd QC toolbox, Lauvset & Tanhua, 2015). After interpolation and prior to gridding, extreme outliers are identified and removed. For each pressure level and Longhurst Biogeographical Province (available at http://comlmaps.org/how-to/layers-and-resources/boundaries/longhurst-biogeographical-provinces/), the interquartile range (IQR) is calculated for each parameter, and values exceeding $1.5 \times$ IQR are flagged as outliers (Johnson & Purkey, 2024). When an outlier is detected for any parameter at a given vertical level within a bin, all variables from that profile and level are discarded. Following Gruber et al., 1998, negative $C_{ant}$ estimates are preserved as negative in the averaging process. Finally, for each grid cell, all valid measurements are averaged to obtain representative values, while cells without observations are left empty. This bin-averaging approach ensures that the gridded products are entirely observation-based and preserve the genuine spatial structure of the compiled dataset.

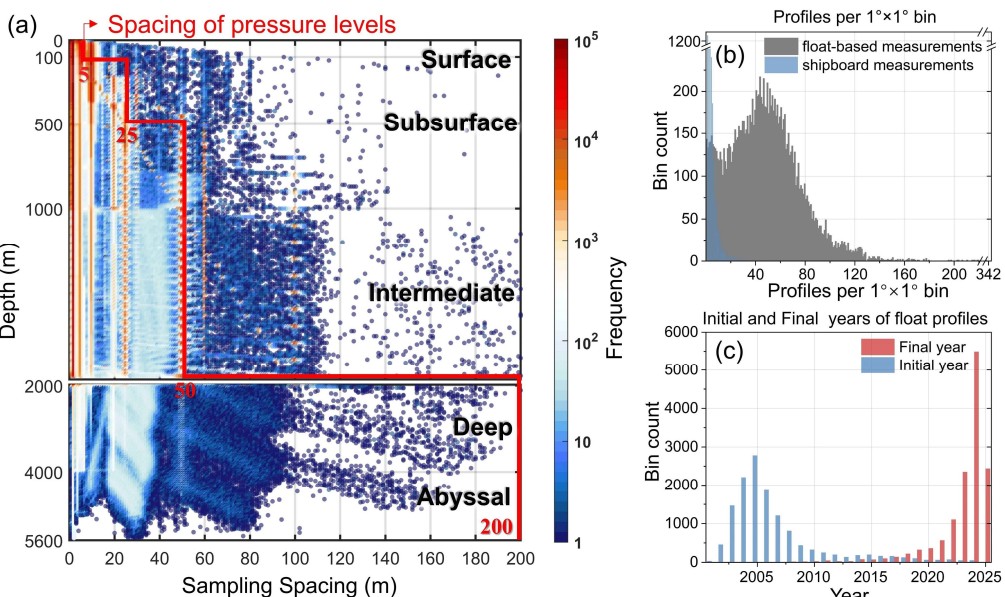

**Figure 3.** (a) Sampling spacing of all Argo floats from sea level to 5600 m. The color of the scattered points shows the frequency. The yellow line illustrates the pressure level used in this study to match sampling spacing and maximum utilize

available profile data. (b) Histogram of number of profiles per $1° \times 1°$ bin. (c) Histogram of initial year (in blue) and final year (in red) per $1° \times 1°$ bin.

### 3.4 Comparative analysis between ship-based observations and float-based reconstructions

To ensure consistency and reliability between float-based reconstructions (including both *full-parameter* and *hydrography-only* pathways) and ship-based observations, three comparative analysis of reconstruction-derived variables ($C_{ant}$ and $\Omega_{ar}$) are implemented.

First, methodological discrepancies between variables from each pathway are assessed using GLODAPv2 data. Specifically, values of $C_{ant}$ and $\Omega_{ar}$ calculated directly from ship-based observations are compared with those estimated from the *full-parameter* pathway ($C_{ant\_ship\_f}$, $\Omega_{ar\_ship\_f}$) and the *hydrography-only* pathway ($C_{ant\_ship\_h}$, $\Omega_{ar\_ship\_h}$), respectively. This analysis quantifies the biases inherent in each reconstruction pathway and is detailed in Section 4.2. Second, within regions exhibiting spatial overlap between float-derived reconstructions and independent ship-based estimates, comparisons of float-based $C_{ant}$ and $\Omega_{ar}$ from each pathway are performed following established cross-over quality control procedure (results shown in Section 4.2). Comparisons are restricted to cases where differences are within $\pm 0.005$ kg m$^{-3}$ in potential density ($\sigma_\theta$), $\pm 0.005$ in neutral density($\tau$), and $\pm 100$ dbar between 1,400 and 2,100 dbar depth range (Bushinsky et al., 2025). This targeted analysis serves to assess potential discrepancies arising from differences between oxygen-equipped floats and floats without oxygen sensors. These two analysis are essential to evaluate potential biases introduced by differences between oxygen-equipped and CTD-only Argo floats, which is particularly important as oxygen-equipped floats comprise approximately 11% of total Argo float deployments in the Southern Ocean—potentially leading to disproportionate representation and biases in gridded products. Finally, a detailed zonal analysis among the gridded products is conducted to evaluate how observational differences impact the spatial consistency and reliability of the final products (results show in Section 4.4).

### 3.5 Uncertainty assessment

### 3.5.1 Uncertainty of oceanic interior carbonate system parameters

Uncertainty assessment in this study is designed to comprehensively quantify error propagation throughout the reconstruction process. The uncertainties of the reconstructed variables—including TA, DIC, pH, and nutrients—are evaluated by considering both the instrument measurement accuracy of Argo sensors and the model-based reconstruction uncertainty. Additionally, estimated $C_{ant}$ and $\Omega_{ar}$ (see Section 3.2) accounts for uncertainty propagation in the calculation (Figure 4).

Instrument measurement errors reflect the inherent limitations of sensors and are quantified either from the specified precision of the instruments or by comparing Argo measurements against independent, high-quality reference data (GLODAP). These measurement errors establish the baseline uncertainty that propagates through subsequent steps.

Model uncertainties are evaluated for each reconstruction method. Different models employ distinct approaches to uncertainty estimation. For example, CANYON-B model expresses neural network weights as probability distributions, providing

probabilistic predictions that incorporate model weight uncertainty. In contrast, ESPER_LIR and ESPER_NN report uncertainties based on the root mean square error (RMSE) of their validation dataset and through interpolation across depth and salinity space.

Additionally, uncertainties in the calculations of $C_{ant}$ and $\Omega_{ar}$ are addressed via Monte Carlo simulation (following the principle described in Qi et al., 2022). Both measurement and model-derived uncertainties are propagated through the TrOCA and CO2SYS calculation steps by repeatedly sampling input variables within their respective uncertainty bounds. This generates distributions of $C_{ant}$ and $\Omega_{ar}$, and the standard deviations of these distribution are taken as the estimate of propagated uncertainty.

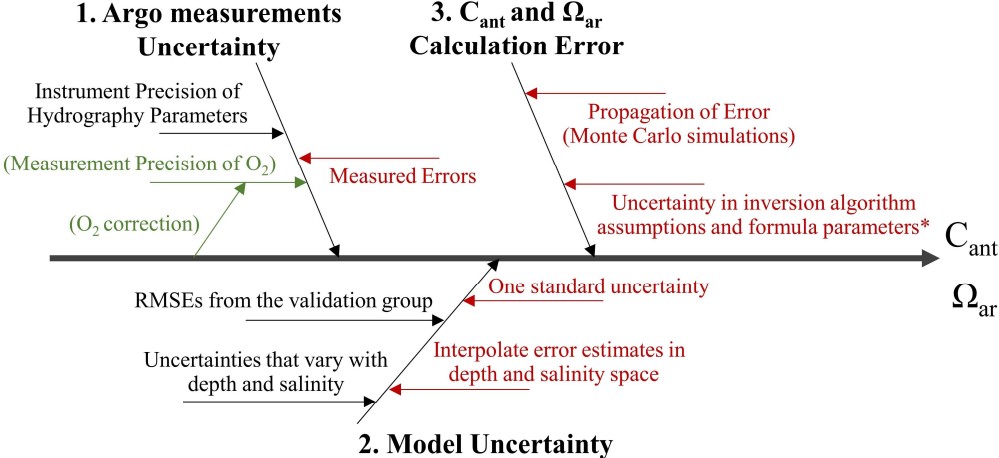

**Figure 4.** Error propagation that may arise during the calculation process of reconstructed variables and calculated values ($C_{ant}$ and $\Omega_{ar}$).

### 3.5.2 Uncertainty of gridded products

The uncertainty estimation for gridded products derived from float-based (and optionally, ship-based) observations consists of two main components: parameter profile sensitivity and spatial spread uncertainty, which together determine the total uncertainty at each grid cell.

At each pressure level, parameter profile sensitivity ($\sigma_{param\_prof}$) is assessed by iteratively perturbing reconstructed variables according to their parameter-specific uncertainties, which is accompanied by the construction of the gridded products as described in Section 3.3. For float-based reconstructions, uncertainties are evaluated as detailed in Section 3.5.1. For ship-based observations, both systematic and random uncertainties are incorporated following recommendations from Carter et al., 2024, with each observations categorized as direct, calculated, or combined; detailed uncertainty estimates are provided in Table A3.

During the gridding process, all N parameter values ($y_{param,i}$) within each 1° × 1° spatial bin and specific pressure level are combined using weighted averaging:

$$y_{grid} = \frac{\sum_{i=1}^{N} w_i \cdot y_{param,i}}{\sum_{i=1}^{N} w_i} \tag{3}$$

where $y_{grid}$ is the gridded parameter value, $y_{param,i}$ is the $i$th profile value interpolated to the pressure level, and $w_i$ is the inverse variance weight (Eq.4), with $\sigma_{param\_prof}$ being the measurement uncertainty of each observation as provided in the SOCOML profile data.

$$w = \frac{1}{\sigma_{param\_prof}^2} \tag{4}$$

The weighted spread of observations within each grid cell ($\sigma_{spread}$) is calculated following Bittig et al., 2018. To estimate the uncertainty of the climatological mean ($\sigma_{grid}$), the $\sigma_{spread}$ is divided by the square root of the effective sample size ($N_{eff}$,) computed using the Kish formula.

$$\sigma_{spread} = \sqrt{\frac{\sum_{i=1}^{N} w_i \cdot (y_{param,i} - y_{grid})^2}{\sum_{i=1}^{N} w_i - \frac{\sum_{i=1}^{N} w_i^2}{\sum_{i=1}^{N} w_i}}} \tag{5}$$

$$\sigma_{grid} = \frac{\sigma_{spread}}{\sqrt{N_{eff}}} \tag{6}$$

$$\sigma_{grid} = \frac{(\sum_{i=1}^{N} w_i)^2}{\sum_{i=1}^{N} w_i^2} \tag{7}$$

## 4 Results and Discussion

### 4.1 Model Performance Comparison

Model performance is evaluated for three widely used approaches for estimating oceanic biogeochemical properties: ESPER_LIR, ESPER_NN, and the CANYON-B. The models are trained on a different version of the GLODAPv2 dataset (CANYON-B on GLODAPv2.2016; ESPER_LIR and ESPER_NN on GLODAPv2.2020). Independent assessment data not included in model training are used for comparison (Table 1). Both the *full-parameter* and *hydrography-only* reconstruction pathways are assessed.

Under the *full-parameter* pathway, ESPER_NN achieves the lowest RMSE for most reconstructed variables (Table 2), including TA (4.37 µmol kg⁻¹), DIC (6.09 µmol kg⁻¹), PO₄ (0.07 µmol kg⁻¹), and SiO₄ (2.59 µmol kg⁻¹). ESPER_LIR performs slightly better for pH and NO3. Generally, ESPER_NN's RMSE values for TA and DIC are several percent lower than those of ESPER_LIR and CANYON-B, demonstrating superior accuracy relative to the observations.

Under the *hydrography-only* pathway, omission of oxygen leads to a notable increase in RMSE for DIC (8.78 µmol kg⁻¹), particularly in deep and abyssal waters (Table 2). This highlights the critical role of oxygen measurements as predictors for deep DIC. Relative to CANYON-B, both TA and DIC exhibit systematic underestimation, with mean full-column mean bias of −0.6 and −1.7 µmol kg⁻¹, respectively. These biases are more pronounced than those reported in earlier evaluation (Carter et al., 2021a), who found biases of −0.4 and −0.8 µmol kg⁻¹ using 2019–2020 GLODAPv2 data. The increasing discrepancies

suggest that prediction errors in CANYON-B may be accumulating over time. Similar trends are observed for ESPER_LIR and ESPER_NN, indicating that periodic updates to model training datasets are necessary to mitigate future underestimation of DIC.

325 Uncertainty magnitudes vary among models, reflecting both differences in model structure and in uncertainty estimation methodologies. CANYON-B, which directly incorporates measurement uncertainties from input variables, produces larger uncertainty. Vertical performance analysis shows that the lowest RMSE values for most variables are found in deep and abyssal layers (comprising 24% of full-column data), while the largest errors are found in surface waters (25%), likely due to greater variability in surface carbonate chemistry.

330 Overall, the ESPER_NN demonstrates the highest accuracy and lowest uncertainty under both reconstruction pathways (Table 2), supporting its selection as the primary model for reconstructing carbonate system parameters and nutrients in the Southern Ocean throughout this work. Based on the estimated RMSE of ESPER_NN, float-derived estimates are expected to fall within twice the model's estimated uncertainty range, serving as a criterion for the quality control applied to the Argo float dataset.

**Table 2.** The statistics (r², RMSE, and mean bias) were compared between estimated parameter values from ESPER_NN and CANYON-B using assessment data added after the original GLODAPv2.2020 release (i.e., all cruises with GLODAPv2 cruise numbers ≥2107, Table 1). In addition, the reconstructed performance of model was examined for the entire water column as well as for specific depth ranges: surface (0–200 dbar), intermediate (1000–2000 dbar), and deep and abyssal (>2000 dbar).

| | | With Oxygen | | | | | | | | | Without Oxygen | | | | | |
| | | ESPER_LIR | | | ESPER_NN | | | CANYONB | | | ESPER_LIR | | | ESPER_NN | | |
| | | RMSE | Bias | Uncertainty | RMSE | Bias | Uncertainty | RMSE | Bias | Uncertainty | RMSE | Bias | Uncertainty | RMSE | Bias | Uncertainty |
|---|---|---|---|---|---|---|---|---|---|---|---|---|---|---|---|---|
| **TA** | All | **4.79** | **0.13** | **4.54** | **4.37** | **0.26** | **4.10** | **4.43** | **-0.6** | **9.56** | **5.11** | **-0.03** | **4.79** | **4.77** | **0.19** | **5.24** |
| | Surface | 5.89 | -0.40 | 6.80 | 4.72 | -0.05 | 5.96 | 4.38 | -0.67 | 9.67 | 5.77 | -0.46 | 6.54 | 4.98 | -0.73 | 8.01 |
| | Intermediate | 4.37 | 0.53 | 3.76 | 4.36 | 0.58 | 3.40 | 4.57 | -0.54 | 9.43 | 4.60 | 0.49 | 4.01 | 4.75 | 0.92 | 4.60 |
| | Deep, Abyssal | 3.42 | 0.22 | 3.26 | 3.30 | 0.29 | 2.96 | 4.12 | -0.98 | 9.88 | 4.92 | -0.39 | 4.30 | 4.39 | 0.71 | 3.73 |
| **DIC** | All | **6.35** | **0.04** | **5.15** | **6.09** | **0.05** | **6.61** | **8.12** | **-1.7** | **15.76** | **7.92** | **-0.25** | **8.91** | **8.78** | **-0.10** | **9.11** |
| | Surface | 10.89 | 1.29 | 8.92 | 10.14 | 0.95 | 11.89 | 12.09 | -2.41 | 29.19 | 12.28 | 2.21 | 16.29 | 12.94 | 4.12 | 16.03 |
| | Intermediate | 3.07 | -0.74 | 3.28 | 3.21 | -0.42 | 4.89 | 5.06 | -1.4 | 9.05 | 3.91 | -0.75 | 4.39 | 5.75 | -1.39 | 5.52 |
| | Deep, Abyssal | 3.27 | -0.29 | 3.48 | 3.34 | 0.20 | 3.40 | 5.44 | -0.95 | 9.04 | 5.52 | -1.22 | 4.98 | 6.37 | -0.99 | 4.16 |
| **pH** | All | **0.018** | **-0.001** | **0.012** | **0.020** | **-0.003** | **0.010** | **0.023** | **-0.003** | **0.019** | **0.024** | **-0.001** | **0.022** | **0.023** | **-0.004** | **0.020** |
| | Surface | 0.030 | -0.005 | 0.017 | 0.035 | -0.006 | 0.017 | 0.039 | -0.01 | 0.020 | 0.039 | -0.008 | 0.040 | 0.039 | -0.011 | 0.035 |
| | Intermediate | 0.010 | -0.001 | 0.008 | 0.009 | -0.001 | 0.006 | 0.011 | -0.002 | 0.018 | 0.011 | -0.001 | 0.011 | 0.011 | -0.001 | 0.011 |
| | Deep, Abyssal | 0.015 | 0.002 | 0.013 | 0.010 | -0.003 | 0.007 | 0.010 | 0.001 | 0.018 | 0.015 | 0.002 | 0.015 | 0.010 | -0.002 | 0.007 |
| **NO₃** | All | **0.92** | **0.02** | **0.74** | **1.08** | **0.01** | **0.55** | **0.98** | **-0.01** | **1.00** | **1.11** | **-0.04** | **1.20** | **0.97** | **-0.01** | **0.96** |
| | Surface | 1.71 | 0.23 | 1.41 | 2.06 | 0.24 | 0.93 | 1.60 | 0.17 | 1.00 | 1.81 | 0.26 | 2.08 | 1.56 | 0.30 | 1.53 |
| | Intermediate | 0.29 | 0.00 | 0.54 | 0.28 | -0.01 | 0.43 | 0.53 | 0.01 | 1.02 | 0.45 | -0.02 | 0.68 | 0.50 | -0.04 | 0.56 |
| | Deep, Abyssal | 0.28 | -0.03 | 0.40 | 0.28 | -0.04 | 0.37 | 0.45 | -0.05 | 1.01 | 0.41 | -0.12 | 0.54 | 0.44 | -0.06 | 0.47 |
| **PO₄** | All | **0.08** | **0.02** | **0.05** | **0.07** | **0.02** | **0.05** | **0.07** | **0.02** | **0.07** | **0.08** | **0.02** | **0.08** | **0.07** | **0.02** | **0.07** |
| | Surface | 0.15 | 0.04 | 0.09 | 0.12 | 0.04 | 0.08 | 0.12 | 0.04 | 0.07 | 0.13 | 0.03 | 0.13 | 0.11 | 0.05 | 0.11 |
| | Intermediate | 0.03 | 0.02 | 0.04 | 0.03 | 0.02 | 0.04 | 0.05 | 0.03 | 0.07 | 0.04 | 0.02 | 0.05 | 0.05 | 0.02 | 0.05 |
| | Deep, Abyssal | 0.02 | 0.01 | 0.03 | 0.02 | 0.01 | 0.03 | 0.04 | 0.01 | 0.07 | 0.03 | 0.01 | 0.04 | 0.03 | 0.01 | 0.04 |
| **SiO₄** | All | **3.12** | **-0.35** | **2.13** | **2.59** | **-0.04** | **1.78** | **3.24** | **-0.07** | **3.32** | **3.77** | **-0.52** | **2.88** | **3.12** | **-0.12** | **2.26** |
| | Surface | 4.25 | 0.58 | 2.89 | 3.43 | 0.50 | 2.66 | 3.60 | 0.52 | 3.09 | 4.39 | 1.06 | 3.20 | 3.99 | 0.60 | 2.82 |
| | Intermediate | 2.67 | -1.07 | 1.58 | 2.12 | -0.52 | 1.30 | 2.90 | -0.32 | 3.42 | 3.47 | -1.21 | 2.26 | 2.39 | -0.26 | 1.78 |
| | Deep, Abyssal | 2.96 | -1.18 | 2.28 | 2.55 | -0.48 | 1.65 | 3.71 | -0.66 | 3.92 | 4.26 | -2.15 | 3.32 | 3.25 | -0.75 | 2.18 |

## 4.2 Evaluation of bias between full-parameter and hydrography-only pathways

We first analyze the methodological discrepancies between the two reconstruction pathways using high-quality shipboard measurements. Biases in $C_{ant}$ and $\Omega_{ar}$ are quantified by using ship-based observations with concurrent $O_2$, TA, and DIC measurements (N = 93,667). Parameter values derived directly from measured shipboard data are compared with those calculated from reconstructed DIC, TA for two pathways (Figure 5).

Under the full-parameter pathway, $C_{ant}$ exhibits a slight negative bias relative to shipboard derived values ($C_{ant\_ship\_M}$), with a median of $-0.08$ μmol kg$^{-1}$ and a mean of $-0.16 \pm 4.74$ μmol kg$^{-1}$ (95 % CI: [$-0.19, -0.12$] μmol kg$^{-1}$). When oxygen is omitted, the $C_{ant}$ bias distribution broadens and shifts slightly positive, with a median of 0.07 μmol kg$^{-1}$ and a mean of $0.02 \pm 6.60$ μmol kg$^{-1}$ (95 % CI: [$-0.02, 0.07$] μmol kg$^{-1}$). $\Omega_{ar}$ biases remain small in both pathways but show a similar pattern: for the *full-parameter* pathway, $\Omega_{ar}$ bias is centered near zero (median = 0.0006; mean = $0.0006 \pm 0.0433$; 95 % CI: [0.0005, 0.0011]), whereas the *hydrography-only* pathway yields a slightly larger mean bias and spread (median = 0.0021; mean = $0.0021 \pm 0.0618$; 95 % CI: [0.0010, 0.0018]). The *hydrography-only* pathway results in a median difference ($C_{ant\_ship\_H} - C_{ant\_ship\_F}$; $\Omega_{ar\_ship\_H} - \Omega_{ar\_ship\_F}$) of $+0.1$ μmol kg$^{-1}$ and 0.001 units with an added methodological uncertainty of $\pm 2.4$ μmol kg$^{-1}$ and $\pm 0.02$ units (Figure A3) compared to the *full-parameter* pathway.

Subsequently, biases in float-based estimates from the two pathways are further assessed. The comparison results, restricted to cases within the 1,400-2,100 dbar depth range and specific seawater property differences (see Section 3.4), are shown in Figure 6a-d. Oxygen concentrations estimated by both pathways exhibit insignificant systematic offset, indicating robust performance of float-based oxygen reconstructions. Biases in float-based $C_{ant}$ estimates exhibit a moderate positive correlation with oxygen biases, especially at higher latitudes, while $\Omega_{ar}$ biases exhibit a slight positive association with $O_2$ biases, particularly in the mid-latitudes. The difference in float-based $C_{ant}$ and $\Omega_{ar}$ between the two reconstructed pathways are within $\pm 10$ μmol kg$^{-1}$ and $\pm 0.075$ unit, respectively. These ranges are roughly half those observed for the bias distributions between float-based and ship-based values.

Overall, the two reconstructed pathways for Argo floats have a narrow bias distribution for reliable Southern Ocean analyses. Considering both the methodological uncertainty and random uncertainty (estimated following Section 3.5.1), the uncertainties of $C_{ant}$ in both pathways are about $\pm 4\sim 6$ μmol kg$^{-1}$, which remain acceptably low for large-scale biogeochemical reconstructions (e.g., compared to Pardo et al., 2014 of $\pm 6$ μmol kg$^{-1}$ and Asselot et al., 2024 of $\pm 5.2$ μmol kg$^{-1}$).

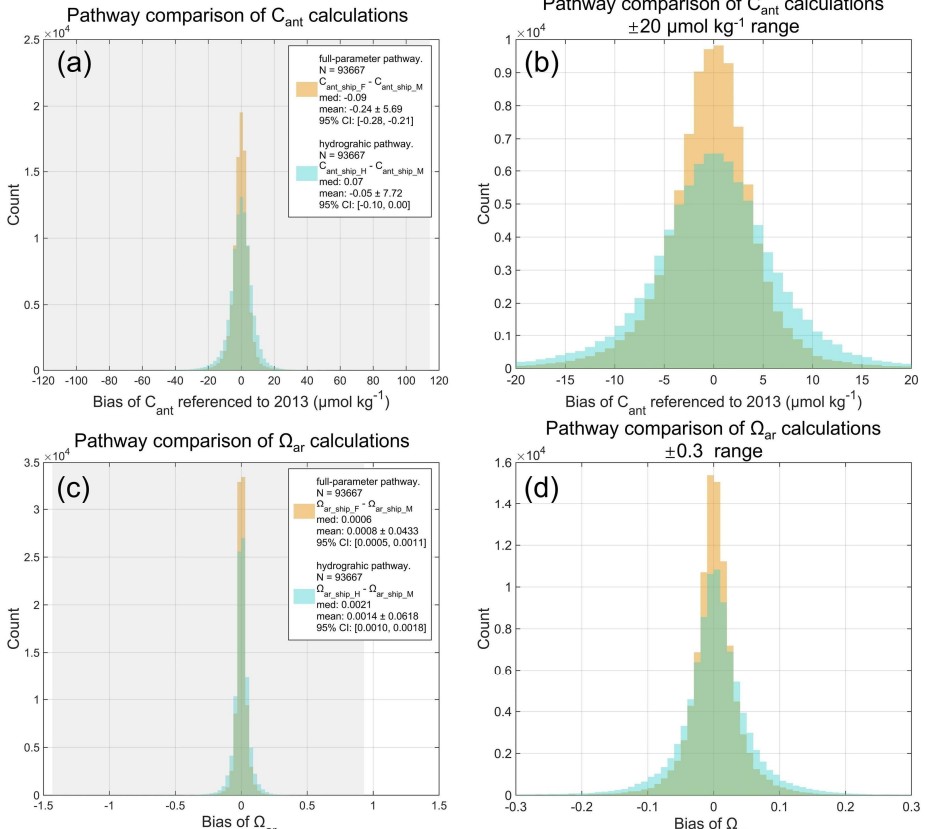

**Figure 5.** Histograms of calculation biases between the *full-parameter* pathway (orange; includes oxygen concentration) and the *hydrography-only* pathway (cyan, excludes oxygen concentration) for (a-b) anthropogenic carbon ($C_{ant}$) and (c-d) aragonite saturation state ($\Omega_{ar}$). Bias is defined as the difference between values calculated using ESPER_NN-derived variables or GLODAP shipboard measurements. The grey background denotes the full range of bias values, with all *x*-axis centered at zero (bias = 0) for visibility. (b) and (d) were the same as (a) and (b), but with a restricted *x*-axis range of ±20 µmol kg⁻¹ for $C_{ant}$ and ±0.3 for $\Omega_{ar}$. Figure legends indicate the calculation pathway, number of data, median values, mean values ± 1 SD, and 95% confidence intervals.

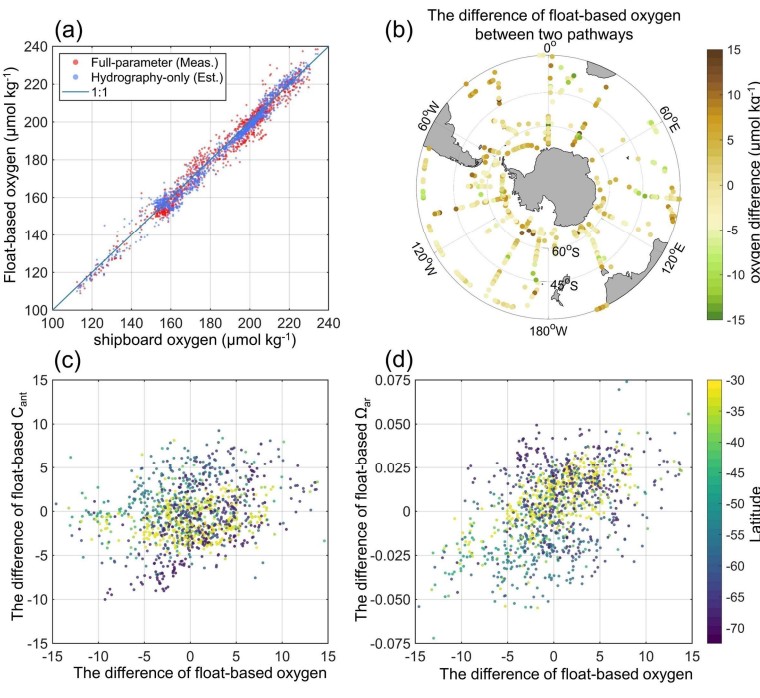

**Figure 6.** Scatter comparisons and spatial distributions of difference in O₂, $C_{ant}$ and $\Omega_{ar}$ between the two reconstruction pathways, restricted to the 1,400-2,100 dbar range. (a) scatter plot comparing float-based and ship-based oxygen measurements

under the *full-parameter* (red) and *hydrography-only* (blue) pathways, and (b) spatial distribution of $O_2$ differences. (c-d) Scatter plots illustrating correlations between differences in $O_2$ and differences in $C_{ant}$ and $\Omega_{ar}$, colored by latitude.

### 4.3 Climatological distributions

Climatological spatial distributions of interior carbonate system parameters are obtained by averaging measured and reconstructed values from both ship-based observations and Argo float-based reconstructions, as well as their calculated values of $C_{ant}$, and $\Omega_{ar}$. Shipboard measurements span 1972-2020, while float–based observations span 2000-2025, with oxygen-equipped floats contributing data since 2003. Figures 7, 8 illustrate the climatological spatial distribution of the interior carbon system parameters of the Float Grid. The spatial patterns of the gridded products are generally consistent, except for $C_{ant}$ and

$\Omega_{ar}$ in the southwestern Atlantic (see Figure B2). To illustrate this discrepancy, the corresponding $O_2$-Float Grid distributions are also provided. Figure 9 presents the distributions in the abyssal layer. Because observations below 4,000 m are extremely scarce, the All-Data Grid is used to provide the most comprehensive depiction of deep-ocean conditions. Section 4.4 further provides a detailed comparison among these products and shipboard estimates.

    The absence of continental barriers across much of the Southern Ocean and the transport of the Antarctic Circumpolar Current

(ACC) result in pronounced meridional gradients dominating the spatial patterns of interior biogeochemical properties. These meridional gradients are closely linked to the spatial distribution of the circumpolar hydrographic fronts, including the Subtropical Front (STF), the Subantarctic Front (SAF), the Polar Front (PF), and the Southern Antarctic Circumpolar Current Front (SACCF), which are indicated by black lines in the climatological distribution maps. The climatological distributions further reflect inter-basin variability driven by ocean basin geometry, bathymetry, and ocean circulation differences among the

Pacific, Atlantic, and Indian Oceans. The averaged profile distributions in the Pacific, Atlantic, and Indian sectors of the Southern Ocean ae shown in Figures 7(d, h) and 8(d, h).

    The distribution of DIC and $C_{ant}$ exhibit strong spatial relationships (Figure 7). In the subsurface layer, high DIC and low $C_{ant}$ concentrations are found south of the PF, the southern boundary of the ACC, due to upwelling of older, DIC-rich and $C_{ant}$-poor deep waters (Marshall & Speer, 2012). Conversely, the northern portion of the Southern Ocean, in north of the SAF,

display low DIC and high $C_{ant}$ concentrations attributed to the transport of Subantarctic Mode Water (SAMW) (Talley, 2013). As depth increases into the intermediate layer (1400-2000 m), $C_{ant}$ concentrations decline significantly, accompanied by increases in DIC. $C_{ant}$ demonstrates pronounced basin-scale variability, with notably low concentrations (0–5 $\mu$mol kg$^{-1}$) in mid-to-high latitudes of the southeastern Pacific Ocean, particularly south of the SACCF between 120°W and 180°W. Conversely, higher $C_{ant}$ concentrations (>20 $\mu$mol kg$^{-1}$) are observed in the Pacific sectors and areas south of the PF in the

eastern Antarctic region. Regions with elevated DIC typically show lower $C_{ant}$ concentrations, and vice versa. In the deep and abyssal layer (2000-5600 m, Figure 9c), the spatial patterns of DIC and $C_{ant}$ remain unchanged, and their vertical profiles flatten. Both DIC and $C_{ant}$ exhibit relatively high concentrations in the eastern Antarctic region, where Antarctic Bottom Waters (AABW) forms (Morrison et al., 2020). This enrichment is consistent with AABW-driven transport of anthropogenic carbon into the deep ocean.

The accumulated $C_{ant}$ uptake and increased DIC concentrations intensify OA, leading to declines in both pH and $\Omega_{ar}$. Spatial distributions of pH and $\Omega_{ar}$ (Figure 8) closely resemble those of DIC (Figure 7a–d). In the subsurface layer, pH exhibits a spatial distribution pattern nearly identical to DIC. However, Figure 8e demonstrates distinctly lower $\Omega_{ar}$ value south of the STF. Both pH and $\Omega_{ar}$ values decrease markedly from the surface to approximately 1000 m, with more gradual declines at depths below 1000 m. In the intermediate and deep layer, the Pacific Sector shows the lowest pH and $\Omega_{ar}$ values, followed by

the Indian Sector and Atlantic Sector, respectively.

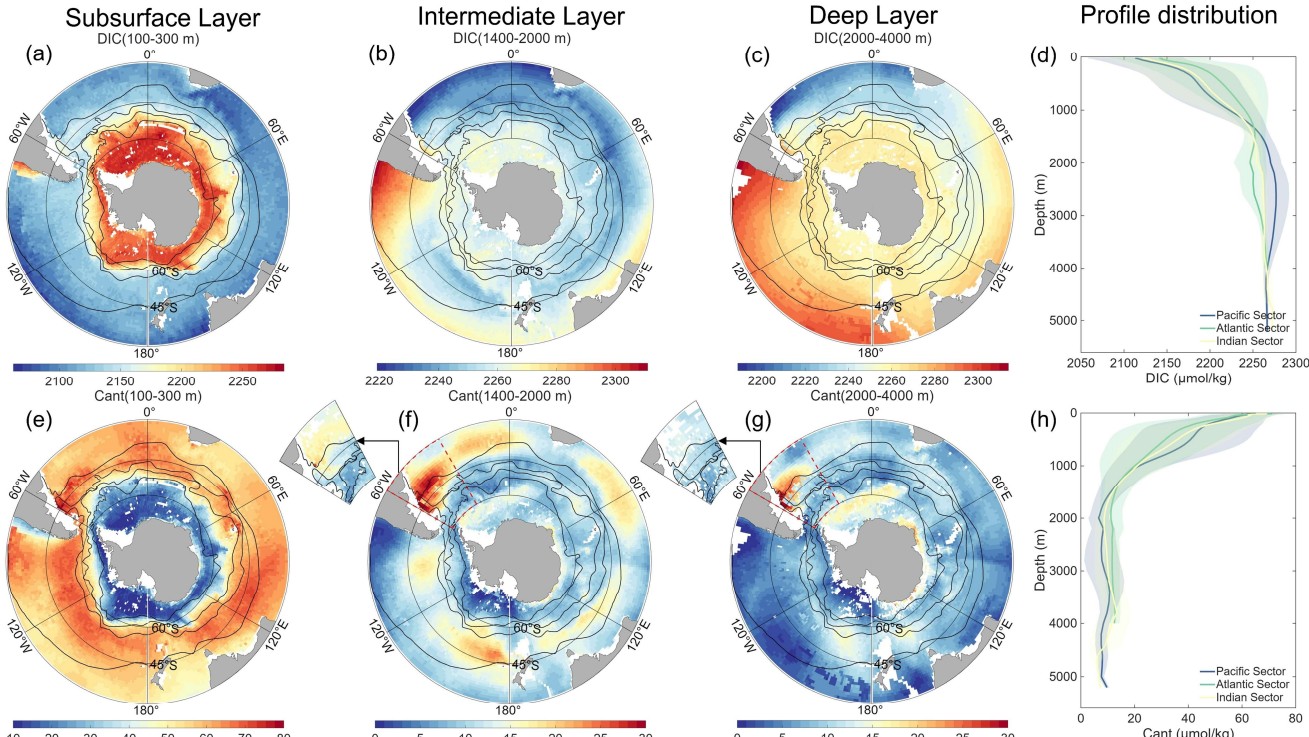

**Figure 7.** Averaged climatological distribution of DIC (a-d) and $C_{ant}$ (e-h) in oceanic sectors and three layers: subsurface layer (100 to 300 m), intermediate layer (1400 to 2000 m), and deep layer (2000 to 4000 m). The climatology is based on the Float Grid derived from Argo profile data spanning 2001-2024 for DIC, while $C_{ant}$ is scaled to the reference year 2013. In the

southwestern Atlantic, where noticeable differences of $C_{ant}$ in deep waters occur between different data products (see explanation in Appendix B2 and B3), the corresponding $O_2$-Float Grid distributions for the intermediate and deep layers are additionally shown as the inset to the subplot (f, g). The thin black lines show, from north to south, the Subtropical Front (STF), the Subantarctic Front (SAF), the Polar Front (PF), and the Southern Antarctic Circumpolar Current Front (SACCF) (Oris et al., 1995). Note that the color scales differ among the individual maps.

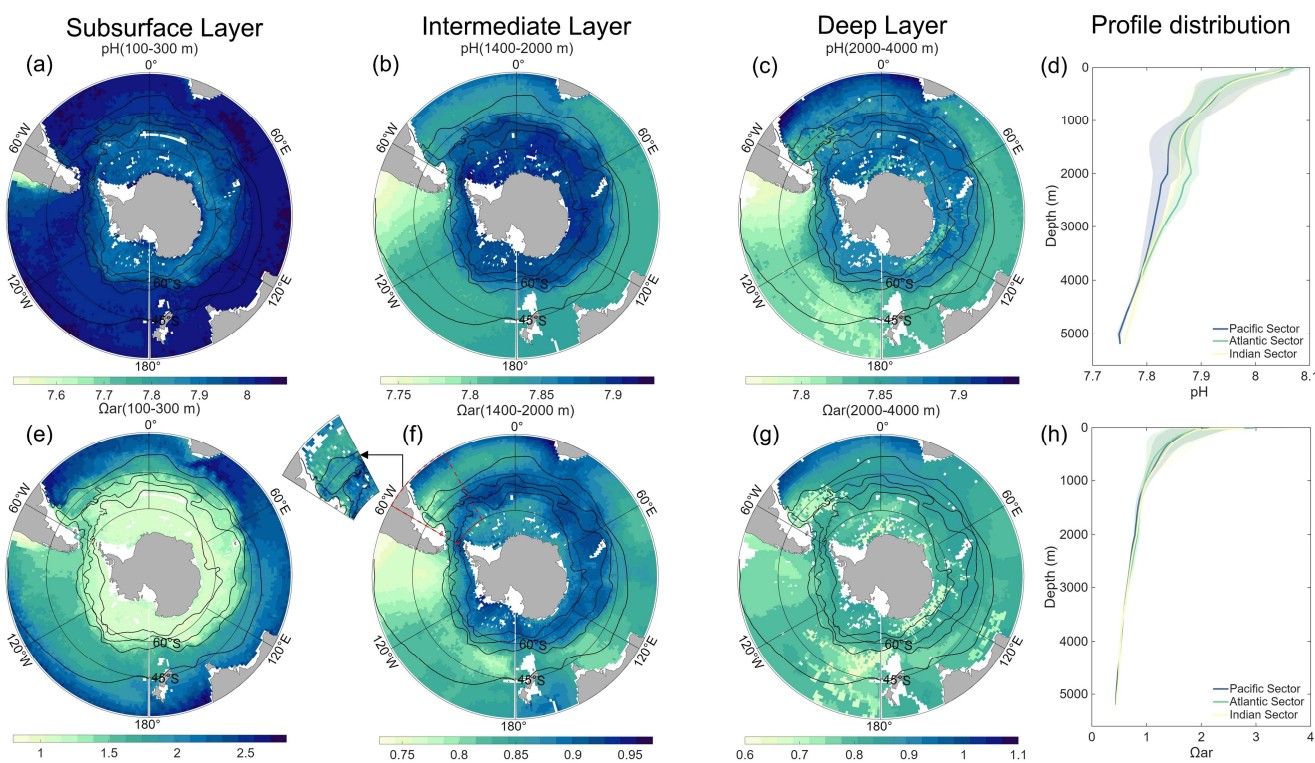

**Figure 8.** Averaged climatological distribution of pH (a-d) and $\Omega_{ar}$ (e-h) in oceanic sectors and three layers: subsurface layer (100 to 300 m), intermediate layer (1400 to 2000 m), and deep layer (2000 to 4000 m). The climatology is based on the Float

Grid derived from Argo profile data spanning 2001-2024. In the southwestern Atlantic, where noticeable differences of $C_{ant}$ in deep waters occur between different data products (see explanation in Appendix B2 and B3), the corresponding $O_2$-Float Grid

distribution for the intermediate layer is additionally shown as the inset to the subplot (f). The thin black lines show, from north to south, the Subtropical Front (STF), the Subantarctic Front (SAF), the Polar Front (PF), and the Southern Antarctic Circumpolar Current Front (SACCF) (Oris et al., 1995). Note that the color scales differ among the individual maps.

## All-Data Grid in Abyssal Layer

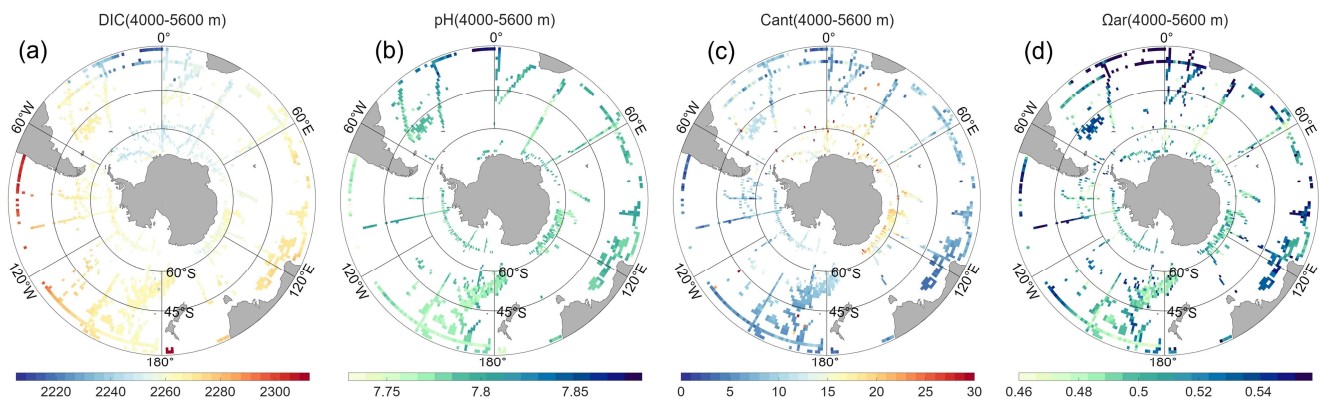

**Figure 9.** Averaged climatological distribution of DIC (a), $C_{ant}$ (b), pH (c), and $\Omega_{ar}$ (d) in the abyssal layer (4000 to 5600 m),
The climatology is based on the All-Data Grid combined float- and ship-based observations. Note that the color scales differ among the individual maps.

## 4.4 Assessment of differences among gridded products

The Float Grid including the Non-$O_2$-Float Grid and the $O_2$-Float Grid represents observation-based climatological products with strong capability to resolve fine-scale horizontal and vertical distributions of interior ocean carbonate system parameters.
Figure 10 presents the latitudinal distributions of $C_{ant}$ and $\Omega_{ar}$ across the Southern Ocean for four gridded products in this study as well as GLODAP-derived data. Additionally, we apply the TRACE method (Carter et al., 2025) to estimate $C_{ant}$, generating a gridded dataset as described in Section 3.3, which serves as an additional comparison (Figure 10b). TRACEv1 adopts a hybrid conceptual framework, with surface-ocean estimates that are observation-based, whereas deep-ocean fields are model-based and tuned against observations.

The four gridded products show latitudinal variations that are closely aligned with the GLODAP-derived data (green lines). Notably, in the intermediate layers characterized by variety in $C_{ant}$ distributions across oceanic basins (Figure 8f), the GLODAP-derived dataset north of 45°S exhibit pronounced zonal gradients, likely due to sparse longitudinal sampling. In contrast, our products, benefiting from enhanced spatial coverage, better capture the integrated regional variations. The TRACE-derived dataset (purple lines) yields lower $C_{ant}$ concentration than the TrOCA-derived values, particularly in
intermediate waters. This difference may partly arise because the model-based approach is difficult to constrain accurately in regions with sparse transient tracer observations and complex vertical structures associated with Southern Ocean upwelling (as suggested by Carter et al., 2025), leading to deviations from the observation-based TrOCA estimates. In deep and abyssal layers, $C_{ant}$ concentrations show an increasing trend from lower latitudes toward higher latitudes (60°S–70°S). This pattern may be linked to the formation of AABW, which drives transport of $C_{ant}$ into the deep oceans (Zhang et al., 2023).

A direct comparison between the $O_2$-Float Grid (red lines) and the Non-$O_2$-Float Grid (black lines) elucidate differences attributable to Core Argo versus BGC Argo observations. Reconstructions of $C_{ant}$ and $\Omega_{ar}$ are broadly consistent across subsurface and intermediate layers for both float types. Significant discrepancies and steep gradients among gridded products are evident in all water layers south of 65°S. Figure A7 illustrates the geographical coverage of float and ship-based observations at high latitudes. In the abyssal layer, float observations are restricted to the eastern Weddell Sea near 4100 m, as
they may not be representative of abyssal-layer conditions. Notably, a hotspot of high $C_{ant}$ and vertically confined low $\Omega_{ar}$ is

identified in the southwestern Atlantic Ocean near the SAF and PF in the Non-O$_2$-Float Grid (Figure 10a). These noticeable differences likely arise from the limited number of shipboard training samples in this region, leading to reduced performance of the machine-learning reconstructions. For regional studies in the southwestern Atlantic, we therefore recommend using the O$_2$-Float Grid provided in this study. The corresponding vertical profiles are presented in Appendix B3. Although float-based

measurements introduce additional uncertainties, their extensive spatial and temporal coverage enables our products to offer unprecedented insight into the previously under-sampled Southern Ocean, particularly in the deep ocean. Overall, the All-Data Grid offers a comprehensive representation of the Southern Ocean interior, while the O$_2$-Float Grid and Non-O$_2$-Float Grid demonstrate the potential and limitations of Argo-based reconstructions for studying carbon dynamics.

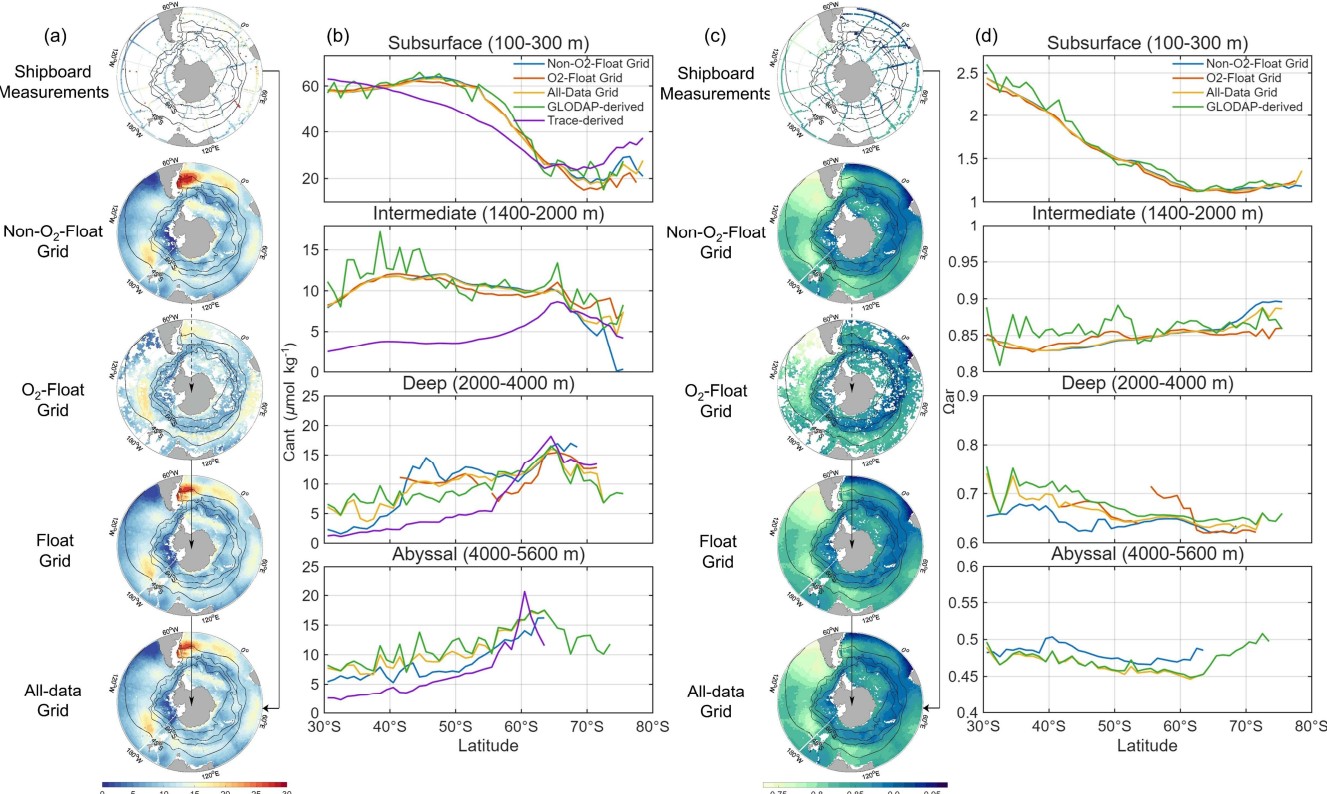

**Figure 10.** Panels (a) and (c) show the climatological distribution of C$_{ant}$ and $\Omega_{ar}$ in the intermediate layers (1400-2000 m), respectively. Panel (b) and (d) present latitudinal distributions of C$_{ant}$ and $\Omega_{ar}$ averaged over four depth layers: subsurface layer (100 to 300 m), intermediate layer (1400 to 2000 m), deep layer (2000 to 4000 m), and abyssal layer (4000 to 5600 m). Black, red, blue, green and purple symbols and lines represent Non-O$_2$-Float Grid, Float Grid, All-Data Grid, GLODAP-derived data, and TraceV1-derived data, respectively.

**4.5 Uncertainties assessment**

The uncertainty of gridded products arise from both the uncertainties in the parameter estimates and mapping (sampling) errors. Considering the nonnegligible trends of accumulated C$_{ant}$, the evaluation of uncertainty in this section mainly focuses on the anthropogenic CO$_2$. Both random errors and potential biases contribute to the uncertainties in the C$_{ant}$ estimates. In Section 4.1, the random errors for individual measurements have been estimated to be about ±4-6 µmol kg$^{-1}$ in the *full-parameter* pathway

and the *hydrography-only* pathway. The potential bias including uncertainty in inversion algorithm assumptions and formula parameters are more difficult to assess quantitatively, but have little effect on the climatological distribution. The mapping errors, reflecting uncertainties introduced during spatial interpolation, are also challenging to evaluate precisely.

Traditionally, distribution maps of carbonate system parameters, including C$_{ant}$, were constructed using limited GLODAPv2 cruises data (sampling stations are shown in Figure 1b), with spatial coverage extended via regression and interpolation

methods (Gruber et al., 2019a; Sabine et al., 2004; Barth et al., 2014). In these earlier products, mapping errors strongly

depended on the vertical and horizontal data distribution and were assumed to be less than 15% Sabine et al., 2004). In contrast, our gridded products leverage the statistical advantage of aggregating multiple independent observations, resulting in gridded uncertainties that are smaller than individual observation uncertainties, and mapping discrepancies reduced to below 7.5 % (Figure A9). Although our approach may underestimate uncertainty due to potential representativity error, our dataset offers a

significant improvement in both accuracy and spatial representativeness over previous gap-filling approaches, and it further extends coverage to the ocean bottom, whereas earlier analyses were largely restricted to the upper 0-3000 m (Gruber et al, 2019). This enhancement is especially valuable for robustly assessing variability and climatological trends in the historically data-sparse Southern Ocean.

## 5 Data availability

The raw Argo profile measurements used in this study are publicly available from the Argo Global Data Assembly Center (GDAC) at /ifremer/argo/dac/. The processed Argo profile dataset and SOCOML gridded products, including the primary product ALL-Data Grid and auxiliary products, are available at https://data.mendeley.com/datasets/xzr59ngmpz/2 (Zhong et al.,2025).

## 6 Conclusions

As the Southern Ocean Argo array has expanded, we applied the ESPER_NN model to reconstruct eight key carbonate system parameters — TA, DIC, pH, $NO_3$, $PO_4$, $SiO_4$, $C_{ant}$ and $\Omega_{ar}$ from Argo profiles. These reconstructions were then gridded into a $1° \times 1°$ product with 84 pressure levels. The input variables were dynamically partitioned into *full-parameter* (with $O_2$ measured) and *hydrography-only* (without $O_2$ measured) pathways to leverage the extensive Argo network. To account for differing data sources, we generated four gridded products: the All-Data Grid, integrating both Argo and GLODAP data, and

the Float Grid, further divided into the Non-$O_2$-Float Grid and the $O_2$-Float Grid. Although the All-Data Grid provides a comprehensive climatological distribution derived from multiple integrated data sources, the Float Grid demonstrates greater internal consistency. This is because discrepancies arising from measurement instrumentation differences cannot be fully eliminated, as clearly illustrated in Figure 7c. Consequently, the All-Data Grid is more suitable for large-scale studies, whereas investigations focusing on smaller regions should incorporate more rigorous analyses of accuracy and uncertainty.

Model comparisons and evaluations reveal increasing underestimation of DIC over time, particularly along the hydrography-only pathway, which lead to progressive underestimation of $C_{ant}$. This variation of bias underscores the inherent constraints of machine learning models trained on data confined to a fixed temporal scope; they cannot extrapolate beyond the observed period to capture emerging trends. Despite this, ESPER_NN maintains robust generalization performance against assessment data. And the bias between two pathways remains relatively small compare to the difference of reconstructed variables and

GLODAP measurements. $C_{ant}$ pathway biases remain within $\pm 10\,\mu mol\,kg^{-1}$, and $\Omega_{ar}$ pathway biases within $\pm 0.075$. These biases exhibit latitudinal variability correlated with oxygen bias. This supports the feasibility of using machine learning models to integrate both Core Argo and BGC Argo data, and highlights the potential for future improvements through the assimilation of nitrate and pH observations from Argo floats.

We offer all gridded products including eight oceanic interior carbonate system parameters, along with their uncertainty

estimates, to the scientific community for advancing Southern Ocean carbon-cycle research and improving new perspective of ocean acidification and carbon sequestration based on observational variables.

**Appendix A: Supplemented tables and figures**

**Table A1**. List of cruises with excluded measurements from the carbonate system internal consistency training dataset presented in this work. Numbers in brackets following recommended adjustment values denote stations removed from the dataset.

| Cruise | Expocode | Recommended adjustment values (+ = add; × = multiply) | | | |
| --- | --- | --- | --- | --- | --- |
| | | TA [+] | DIC [+] | PO$_4$ [×] | SiO$_4$ [×] |
| 2 | 06AQ19860627 | | 12 | | |
| 236 | 316N19720718 | -12 [24-61] | | | |
| 240 | 316N19831113 | | | 0.88 | |
| 297 | 323019940104 | -12 | | | |
| 378 | 35MF19990104 | | 43 | | |
| 430 | 49HG19950414 | | -777$_c$ | | |
| 441 | 49HH19941213 | -16 | | | |
| 696 | 74DI20041103 | | | | 0.89 |
| 718 | 90MS19811009 | -12 | | | |

-777=Poor data, no adjustment suggested. If one of the three carbon system parameters—DIC, TA, or pH—is calculated, it is annotated with a subscript c.

**Table A2**. All GLODAPv2 cruise located in Southern Ocean that have pH values.

| | Expocodes | Note |
| --- | --- | --- |
| 1 Pure spectrophotometric measurements | 29HE20130320, 320620140320, 33RO20161119, 33RR20160208, 320620170703, 320620170820, 320620180309, 325020190403, 33RO20180423 | Used. |
| 2.1 Impure spectrophotometric measurements with adjustment to calculations from TA and DIC | 318M20091121, 31DS19960105, 33RO20071215, 33RO20110926, 33RR20080204, 35MF20080207, 49NZ20030803, 49NZ20071122 | Used. (Carter et al., 2018) |
| | 33RO20131223 | Used. |
| 2.2 Impure spectrophotometric measurements with adjustment applied to submitted data | 320620110219, 33RO20100308, 33RR20090320, 49NZ20121128, 49NZ20130106, 29HE20190406 | Used. |
| 2.3 Impure spectrophotometric with un-calculate-able pH | 29HE20010305, 29HE20020304, 29HE20100208, 33RO20050111, 90AV20041104 | Not used. |
| 3 Calculations | 323019940104, 33MW19950922, 09AR20141205, 49NZ20191229, 74JC20181103, 740H20111224, 74EQ20101018, 74EQ20191202 | Used. |
| 4 Potentiometric measurements | 35A319950113 | Not used. |

All TA and DIC data from GLODAP used in this study is measured. (Carter et al., 2024).

**Table A3**. Uncertainty estimation for measurements and calculations from GLODAP

| | | TA | DIC | pH | NO$_3$ | PO$_4$ | SiO$_4$ |
| --- | --- | --- | --- | --- | --- | --- | --- |
| GLODAP Measurements | Pure spectrophotometric pH | 2 | 2 | 0.006 | 2% | 2% | 2% |
| | Impure spectrophotometric pH | | | 0.010 | | | |

| Calculations | - | - | 0.008 | - |

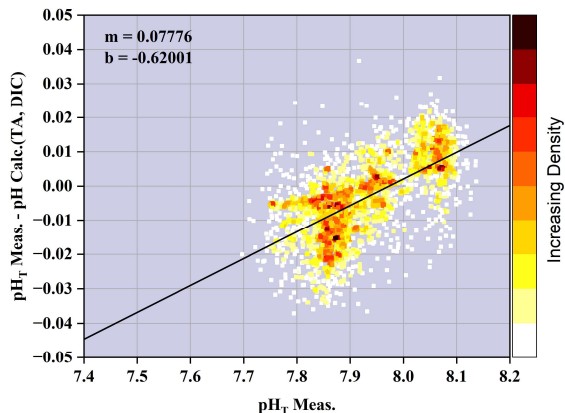

**Figure A1**. NO.1042 cruise's (Expocode: 33RO20131223) scatter plot and linear fitting of measured pH and discrepancy between measured and calculated pH.

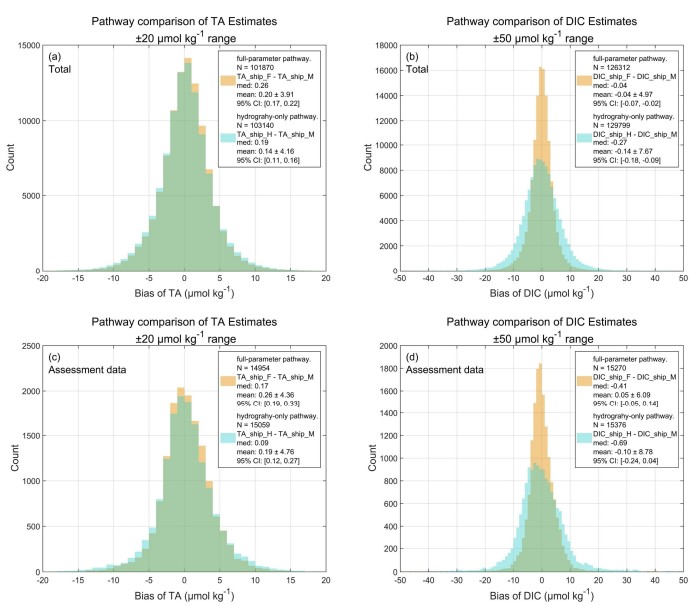


**Figure A2**. Histograms of biases between the full-parameter pathway (orange; includes oxygen concentration) and the *hydrography-only* pathway (cyan, excludes oxygen concentration) for TA and DIC. Bias is defined as the difference between values calculated using ESPER_NN-derived variables or GLODAP shipboard measurements. The *x*-axis was restricted within a range of $\pm 20$ μmol kg$^{-1}$ for TA and $\pm 50$ μmol kg$^{-1}$ for DIC. (a-b) Based on total data of GLODAP; (c-d) Based on assessment

data of GLODAP. Figure legends indicate the calculation pathway, number of data, median values, mean values $\pm$ 1 SD, and 95% confidence intervals.

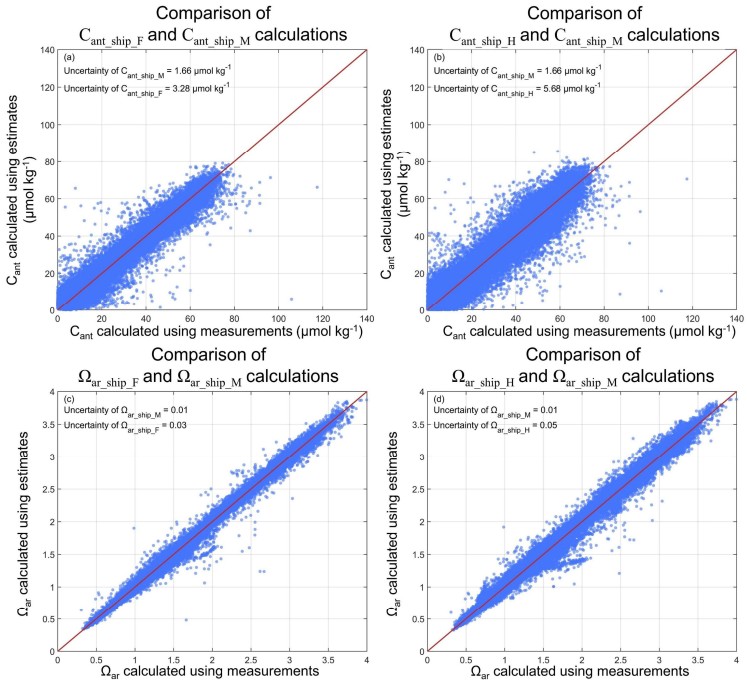

**Figure A3**. Intercomparison between $C_{ant}$ and $\Omega_{ar}$ calculations based on ESPER_NN-derived variables and direct measurements. (a-b) Scatterplots of $C_{ant}$ concentration calculated using ESPER_NN-derived variables and direct shipboard measurements. (c-d) Same as (a-b), but for $\Omega_{ar}$ values. The uncertainties are showed in the left-top of subplot a-d.

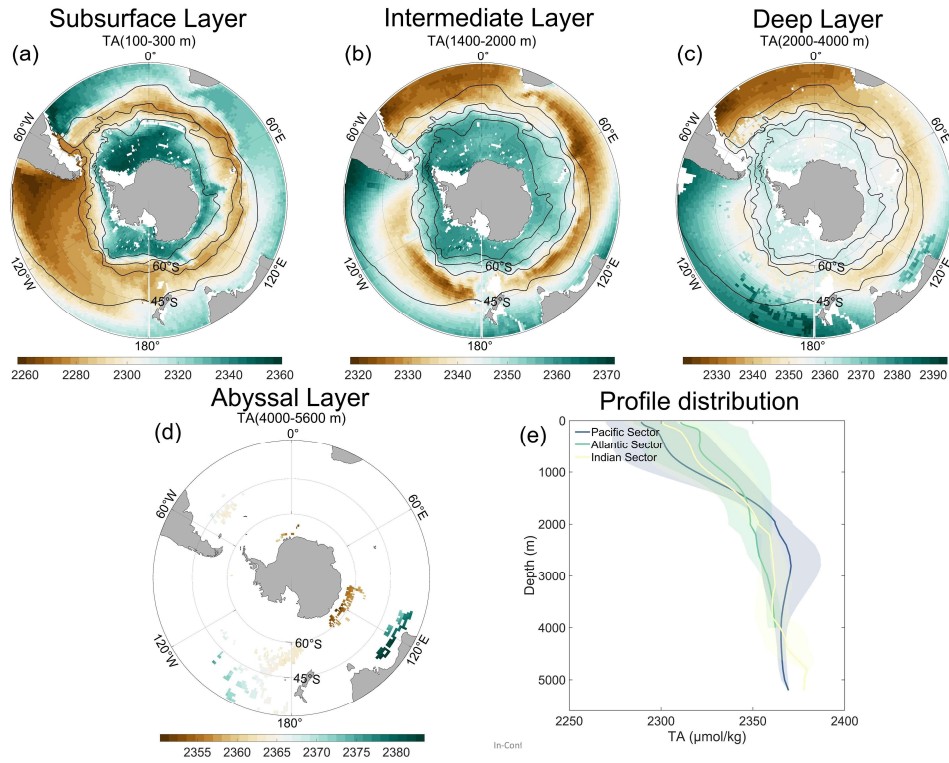

**Figure A4.** Averaged climatological distribution of TA (a-d) in oceanic sectors and three layers: subsurface layer (100 to 300 m), intermediate layer (1400 to 2000 m), deep layer (2000 to 4000 m). The climatology is based on the Float Grid derived from Argo profile data spanning 2001-2024. The thin black lines show, from north to south, the Subtropical Front (STF), the Subantarctic Front (SAF), the Polar Front (PF), and the Southern Antarctic Circumpolar Current Front (SACCF) (Oris et al., 1995). Note that the color scales differ among the individual maps.

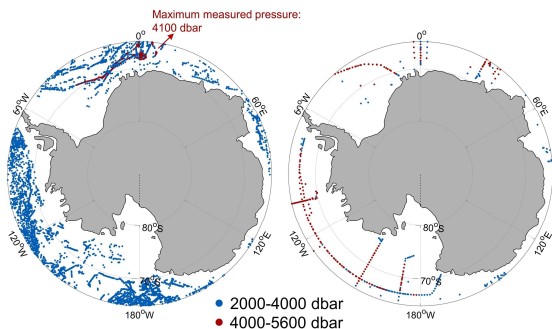

**Figure A5.** Averaged climatological distribution of $NO_3$ (a-d), $PO_4$ (e-h), and $SiO_4$ (i-l) in oceanic sectors (m-o) and four layers: subsurface layer (100 to 300 m), intermediate layer (1400 to 2000 m), deep layer (2000 to 4000 m), and abyssal layer (4000 to 5600 m). The climatology is based on the Float Grid derived from Argo profile data spanning 2001-2024. The thin black lines show, from north to south, the Subtropical Front (STF), the Subantarctic Front (SAF), the Polar Front (PF), and the Southern Antarctic Circumpolar Current Front (SACCF) (Oris et al., 1995). Note that the color scales differ among the individual maps.

**Figure A6.** The location of the observations measured by Argo floats (a) or ship (b) in south of 65°S. The blue and red symbol denotes measurement in the intermediate layer (2000-4000 dbar) and the abyssal layer (4000-5600 dbar), respectively.

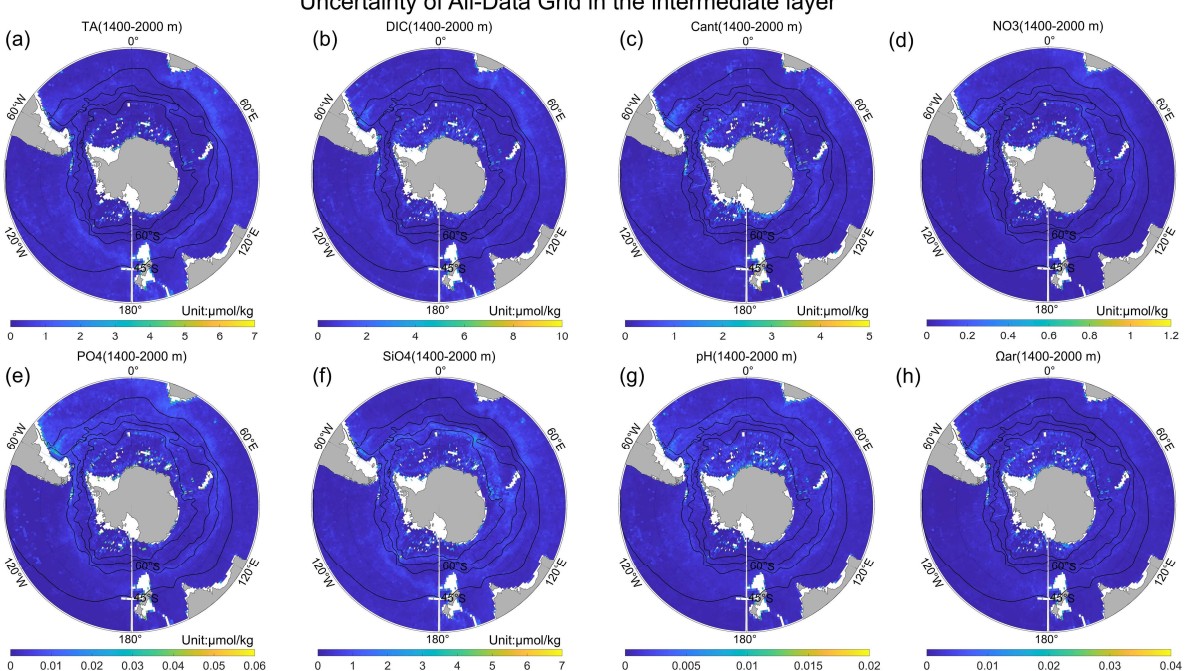

**Figure A7.** The uncertainty of the All-Data Grid in the intermediate layer (1400-2000 m) for (a) TA, (b) DIC, (c) $C_{ant}$, (d) $NO_3$, (e) $PO_4$, (f) $SiO_4$, (g) pH, and (h) $\Omega_{ar}$.

560

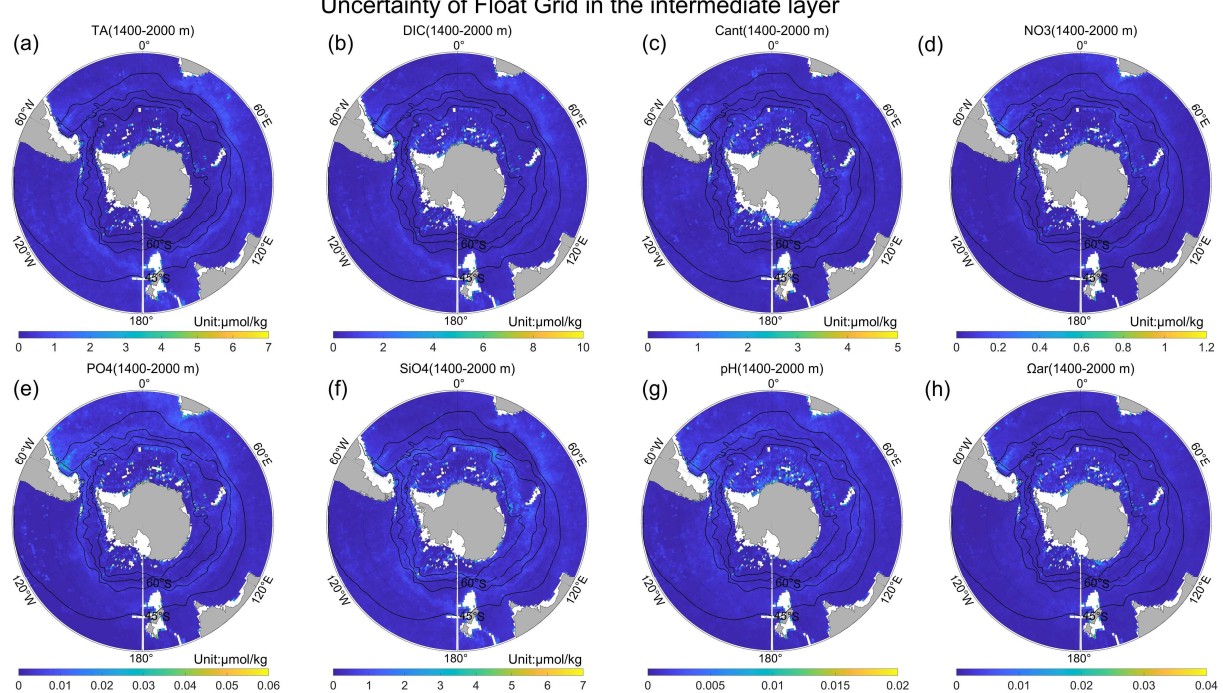

**Figure A8.** The uncertainty of the Float Grid in the intermediate layer (1400-2000 m) for (a) TA, (b) DIC, (c) $C_{ant}$, (d) $NO_3$, (e) $PO_4$, (f) $SiO_4$, (g) pH, and (h) $\Omega_{ar}$.

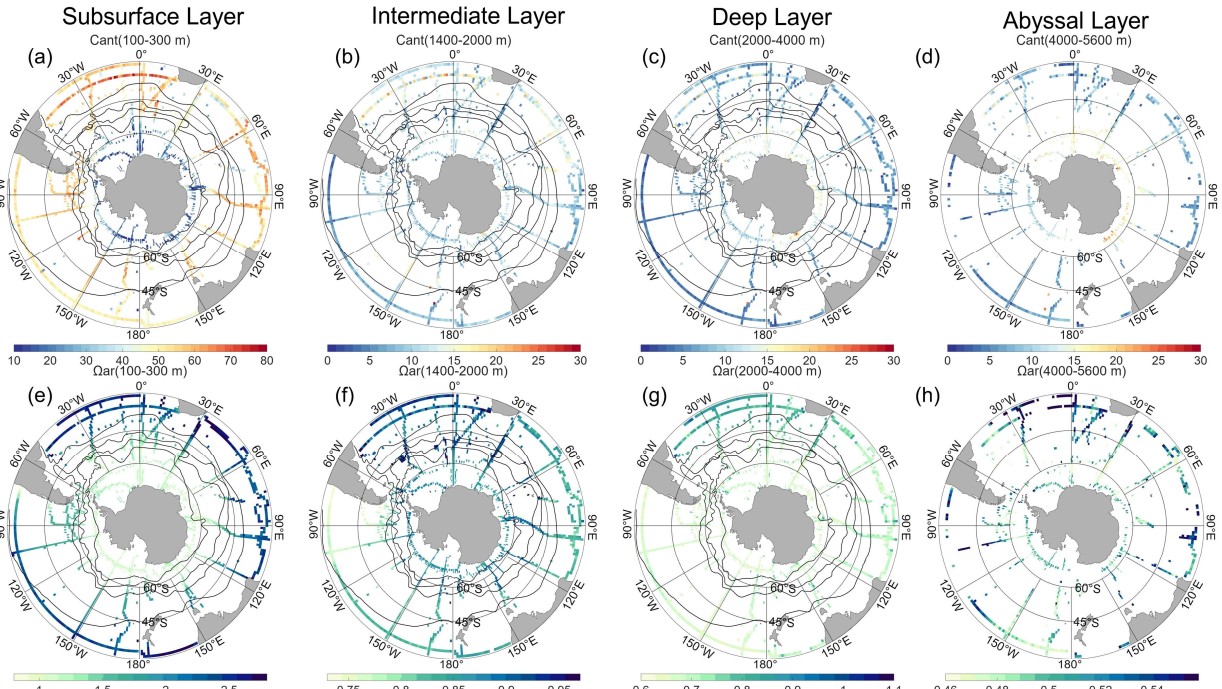

**Figure A9.** Averaged climatological distribution of $C_{ant}$ (a-d) and $\Omega_{ar}$ (e-h) in oceanic sectors and three layers: subsurface layer (100 to 300 m), intermediate layer (1400 to 2000 m), deep layer (2000 to 4000 m), and abyssal layer (4000 to 5600 m). The climatology is derived from GLODAPv2.2023, while $C_{ant}$ is scaled to the reference year 2013. The thin black lines show, from north to south, the Subtropical Front (STF), the Subantarctic Front (SAF), the Polar Front (PF), and the Southern Antarctic Circumpolar Current Front (SACCF) (Oris et al., 1995). Note that the color scales differ among the individual maps.

## Appendix B: Supplement to the methods

### B1 Scaling method

To scale the anthropogenic $CO_2$ concentration, we follow Gruber et al., 2019a to estimate the scaling ratio $\alpha$ of the changes between the periods of $t_1$(2001) and $t_2$(2024) relative to the preindustrial $t_0$(1750):

$$\alpha = \frac{\Delta_t pCO_2^{atm}(t_2 - t_1)}{\Delta_t pCO_2^{atm}(t_0 - t_1)} \cdot \frac{\gamma(t_0..t_1)}{\gamma(t_1..t_2)} \cdot \frac{\xi(t_1..t_2)}{\xi(t_0..t_1)}$$

where $\alpha$ depends mainly on the ratio of the change in atmospheric $CO_2$ ($pCO_2^{atm}$), but is modified by the changes in the revelle factors ($\gamma$) and changes in the air-sea disequilibrium ($\xi$).

Using $pCO_2^{atm}$ = 280 ppm for $t_0$, 371 ppm for $t_1$, and 423 ppm for $t_2$ (Lan et al., 2025), the ratio of the changes in $pCO_2^{atm}$ is 0.57 with a very small uncertainty of about ±0.01 considering round up. Taking the revelle factor for 1950 for $\gamma(t_0..t_1)$ and that for 2013 for $\gamma(t_1..t_2)$ yields a ratio $\gamma(t_0..t_1)/\gamma(t_1..t_2)$ of 0.90±0.02 for the Southern Ocean (south of 30°S). These revelle factors were derived by using products from Gregor & Gruber, 2021. Considering the trends of decrease of the air-sea equilibrium changes relatively small in subtropics and high latitudes (Matsumoto & Gruber, 2005), we use 0.94±0.05 for the ratio $\xi(t_1..t_2)/\xi(t_0..t_1)$ following Gruber et al., 2019a. Using all ratio values, $\alpha$ are set as 0.48±0.04 (0.019986 $yr^{-1}$). Assuming the ocean reaches the constant steady state over 2001-2024, $C_{ant}$ is normalized as referring to scaling equations of Carter et al., 2021a:

$$C_{ant}(t^{ref}) = C_{ant}(t) \cdot e^{0.019986 \cdot (t^{ref} - t)}$$

where $C_{ant}(t^{ref})$ is the normalized $C_{ant}$ concentration at the reference year $t^{ref}$ (set as 2013, the median Argo observation year), and $C_{ant}(t)$ is the estimates for year t.

Figure B1 shows the climatological distribution of $C_{ant}$ is insensitive to uncertainty in the scaling factor, with anomalous change remaining within ±1 μmol kg$^{-1}$. A smaller scaling factor (α=0.44, Figure B1a-d) produces slightly higher $C_{ant}$ values, whereas a larger factor (α=0.52, Figure B1e-f) yields lower values, consistent with different assumed rates of oceanic $CO_2$ accumulation.

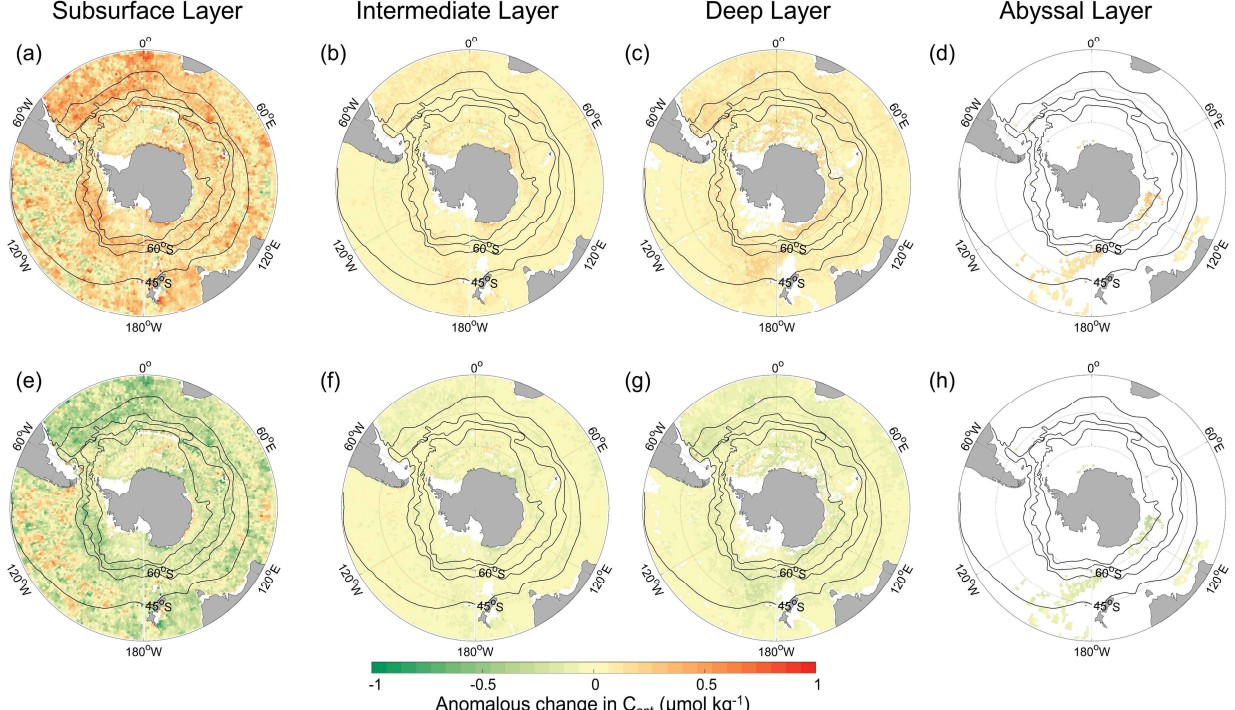


**Figure B1.** Sensitivity of anomalous change in $C_{ant}$ distribution to the value of the scaling factor α=0.44 (a-d) and α=0.52 (e-h) in four layers: subsurface layer (100 to 300 m), intermediate layer (1400 to 2000 m), deep layer (2000 to 4000 m), and abyssal layer (4000 to 5600 m).

**B2 Differences between reconstruction pathways for float-based products**

To quantify the differences between the two reconstruction pathways of Argo floats, BGC-Argo floats equipped with $O_2$ sensors are used to compare the hydrography-only pathway with the full-parameter pathway. For each 1° × 1° grid cell, the averaged vertical difference in $C_{ant}$ and $\Omega_{ar}$ between the two pathways are calculated (Figure B2). Because the full-parameter pathway incorporates measured $O_2$ and therefore provides better-constrained results, its reconstructions of $C_{ant}$ and $\Omega_{ar}$ are regarded as more accurate. The comparison reveals a clear overestimation of $C_{ant}$ and a localized underestimation of $\Omega_{ar}$ in the

southwestern Atlantic under the hydrography-only pathway. The $\Omega_{ar}$ difference primarily occurs within the intermediate layer. Similar but weaker differences are observed near the Ross Sea (overestimated $C_{ant}$, underestimated $\Omega_{ar}$) and along the Pacific coast of South America (underestimated $C_{ant}$, overestimated $\Omega_{ar}$), although their spatial patterns are much less pronounced. The pronounced differences in the southwestern Atlantic likely arises from the limited number of shipboard training samples in this region (only about 1.3 % of the quality-controlled GLODAPv2.2023 measurements in the Southern Ocean contain both

TA and DIC measurements). The performance of the machine-learning reconstructions and the gridded dataset is expected to improve as additional high-quality data become available.

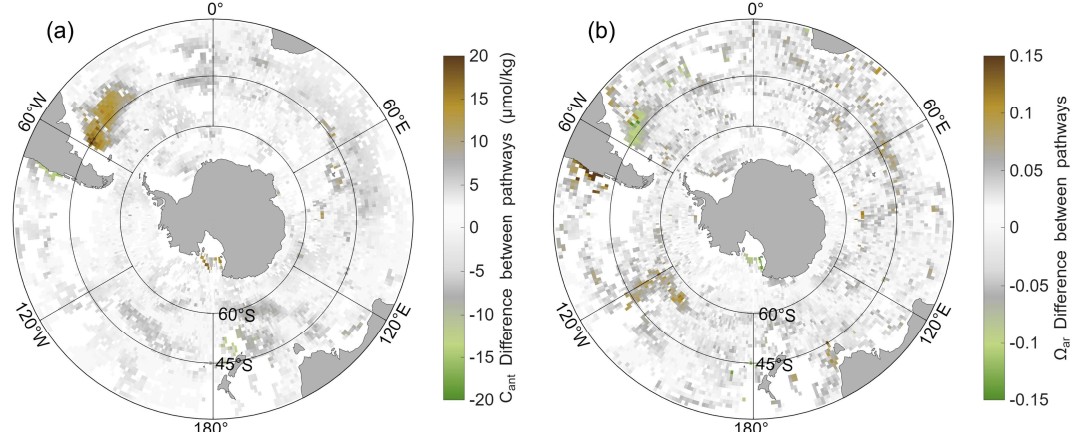

**Figure B2.** Differences between reconstruction pathways for (a) $C_{ant}$ (μmol kg⁻¹) and (b) $\Omega_{ar}$ from BGC-Argo floats with $O_2$ sensors. For each 1° × 1° bin, the mean vertical difference between the hydrography-only and full-parameter pathways are calculated. Values falling within the range derived from float–shipboard matchups are shown in grey, whereas brown and green indicate positive and negative differences, respectively.

**B3 Regional profile distribution in the southwestern Atlantic**

From the climatological distribution of $C_{ant}$ and $\Omega_{ar}$ in the $O_2$-float Grid and non-$O_2$-float Grid, a hotspot of high $C_{ant}$ concentrations and low $\Omega_{ar}$ appears in the southwestern Atlantic Ocean (SW Atlantic) near the SAF and PF. Temperature and salinity profiles from Core and BGC Argo floats are generally consistent, and the profile distributions of $O_2$ measured by BGC Argo floats agree well with those reconstructed from Core Argo data (Figure B3), confirming that the $O_2$ reconstruction in this region is reliable. However, the profiles of TA, DIC, and consequently $C_{ant}$ and $\Omega_{ar}$ exhibit noticeable differences (Figures B4), arising from the inherent limitations of the machine-learning model that lead to distinct reconstruction pathways (detailed in Appendix B2). In the SW Atlantic, $C_{ant}$ exhibits a more stratified vertical structure rather than the deep-penetrating signal captured by the non-$O_2$-Float grid. For regional studies, particularly in the southwestern Atlantic, we recommend using the $O_2$-Float Grid provided in this study.

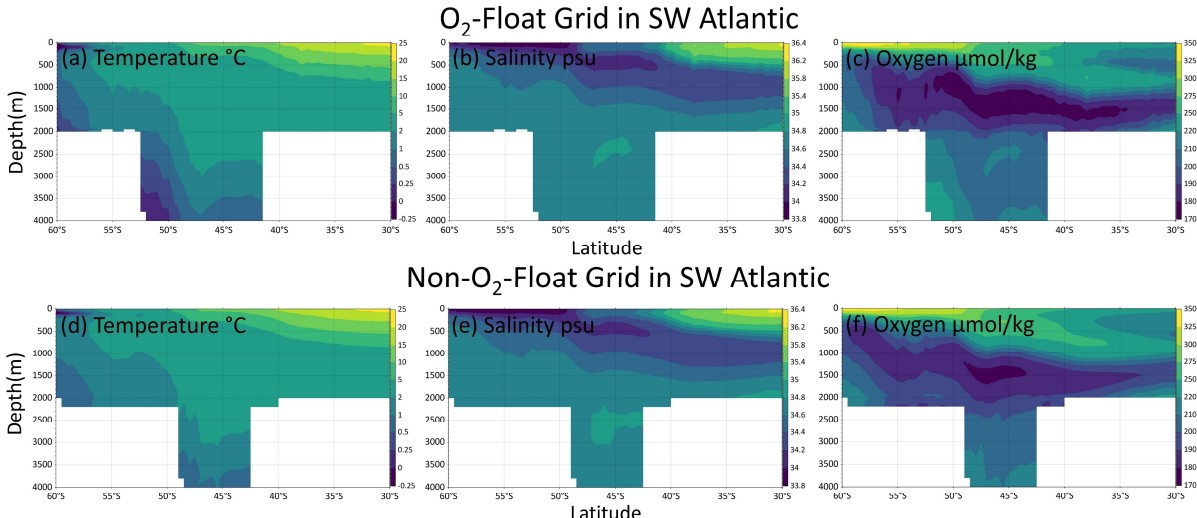

**Figure B3.** Zonal-mean sections (averaged between 60°W and 30°W) of temperature, salinity, and oxygen from 60°S to 30°S in the southwestern Atlantic Ocean. Panels (a-c) show profiles from the $O_2$-Float Grid, and panels (d-f) from the Non-$O_2$-Float Grid. Note that oxygen is measured in panel (c) and reconstructed in panel (f).

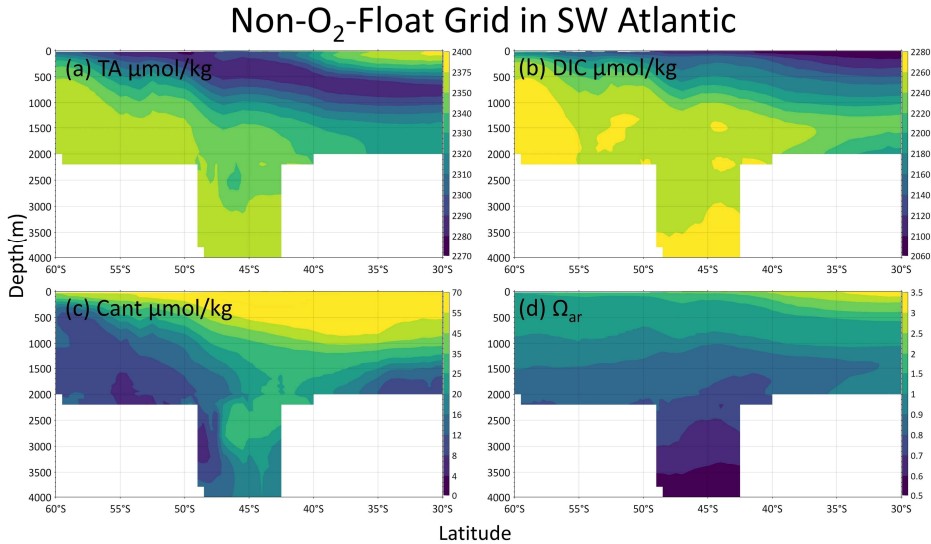

## O$_2$-Float Grid in SW Atlantic

**Figure B4.** Zonal-mean sections (averaged between 60°W and 30°W) of TA (a), DIC (b), C$_{ant}$ (c), and $\Omega_{ar}$ (d) from 60°S to 30°S in the southwestern Atlantic Ocean. The profiles are from the O$_2$-Float Grid.

## Non-O$_2$-Float Grid in SW Atlantic

**Figure B5.** Zonal-mean sections (averaged between 60°W and 30°W) of TA (a), DIC (b), C$_{ant}$ (c), and $\Omega_{ar}$ (d) from 60°S to 30°S in the southwestern Atlantic Ocean. The profiles are from the Non-O$_2$-Float Grid.

## Author contributions

Conceptualization, D.Q. and Y.W.; Data curation: W.Z; Methodology, W.Z., Y.W., and C.L.; Resources, X.M., D.Q., and W.G.; Writing original draft preparation, W.Z; Writing review and editing, all authors.

## Competing interests

The authors declare that they have no conflict of interest.

## Acknowledgement

This work was supported by the Ocean Negative Carbon Emissions (ONCE) Program. We thank the many contributors to the datasets of GLODAP and Argo Program. This work was funded by the Independent Research Projects of Southern Marine Science and Engineering Guangdong Laboratory (Zhuhai) (SML2021SP306), National Natural Science Foundation of China

(Grant 42576268 and 42171464), Fujian Natural Science Foundation (Grant 2025J09045), 2024 International Cooperation Seed Funding Project for China's Ocean Decade Actions (Grant GHZZ370284000202402000028), Hubei Provincial Science and Technology Program Project (2025BEB017), and the Fundamental Research Funds for the Central Universities (Grant ZNJC202415 and 413000028). D.Q. was supported by the National Youth Talent Program of China and the Special

Professorship of the National Major Talent Engineering of China.

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
