# Peer review of "High-resolution spatiotemporal fields of Southern Ocean interior carbonate system parameters integrated from float- and ship-based observations"

_Earth System Science Data, 2025_

## Referee Comment (RC1)

Review of the Manuscript Number: essd-2025-473: Title: High-resolution spatiotemporal fields of Southern Ocean interior carbonate system parameters integrated from float- and ship-based observations by Wanqin Zhong et al.

See attached document.

Review invitation: 14/9/2025
Review accepted: 14/9/2025
Review sent: 21/9/2025

;;;;;;; General comment

In this paper authors present a new dataset of the carbonates system, called SOCOML (Southern Ocean CO2 Machine Learning), derived from shipboard (GLODAP) and float (Argo) data in the Southern Ocean (here south of 30°S). This is a climatological product that includes aragonite saturation state ($\Omega$ar) and anthropogenic CO2 concentrations (Cant) for a reference year (2013). Such climatology (for DIC and TA) has been previously developed, based on shipboard data (e.g., Lauvset et al, 2016; Keppler et al 2020, 2023). Like for surface pCO2 and air-sea CO2 fluxes products (e.g. SOCOM project, Rödenbeck et al, 2015) successfully used to constraint the estimate of the global carbon budget (Friedlingstein et al, 2025) it is important that different climatology for the ocean interior are available for the community (see also a list of such products in Jiang et al, 2025).

After a clear introduction, authors describe the methodology and uncertainties. Three gridded products are offered (with or without O2) at 1x1 degree resolution and 84 levels in the water column down to the abyssal domain. This is especially important in the Southern Ocean where AABW are formed and contain significant Cant concentrations (e.g. Rios et al, 2012; Pardo et al, 2017; Mahieu et al, 2020; Zhang et al 2023). This is also challenging as previous analyses explored the changes of Cant inventories over 0-3000m (Gruber et al, 2019; Müller et al, 2023). Here the product extend to the bottom.

The reconstructed fields for AT, DIC, O2 and nutrients are used to calculate $\Omega$ar and Cant concentrations (here derived from TrOCA method). I suspect this new data-sets could be easily used to estimate Cant using other methods. The results (climatology of TA, DIC, and Cant concentrations) are also important to validate OBGM and ESM models that suffer to reproduce seasonal to multi-decadal DIC and Cant variability. The results will be also probably useful for paleo-oceanography studies (e.g. using pre-industrial DIC profiles deduced from the data presented in this work when calculate DIC-Cant).

Interestingly, through a comparison of different results, authors indicate that the bias in DIC and Cant depend on the dataset used for the training, highlighting the need of maintaining regular update of the dataset such as GLODAP. This is especially true in region where data are relatively sparse such as near coastal Antarctica and in the seasonal ice zone during austral winter. By the way, I was wondering how the sea-ice is taken into account in the method (or if it should be taken into account).

The manuscript is clear, tables and figures adapted. The dataset will be certainly useful for many studies including validation of models. I recommend publication after some clarifications. Below are listed specific and minor comments.

;;;;;;;;;;; Specific comments:

C-01: Title: "High-resolution spatiotemporal fields of Southern Ocean interior carbonate system parameters integrated from float- and ship-based observations". The products are at spatial scale of

1 x 1 degre (not really high-resolution) for a reference year and include anthropogenic CO2 (Cant) so I would suggest change the title: "Climatological fields of Southern Ocean interior carbonate system parameters and anthropogenic CO2 integrated from float- and ship-based observations".

C-02: Line 68: « while the Global Ocean Data Analysis Project version 2 (GLODAPv2, Olsen et al., 2016) offers quality-controlled data product from the surface into the ocean interior includes TA and DIC." It also offered climatology of TA, DIC in the interior ocean (Lauvset et al, 2016).

C-03: Line 70: Maybe also refer here to the last RECCAP-2 story for the SO (Hauck et al, 2023).

C-04: Line 99: "TA, DIC, pH (total scale), nitrate (NO3), phosphate (PO4) and silicate (SiO4) are obtained through neural networks". Add also O2 in this list as this is used for Cant calculation ?

C-05: Line 100-101: « And Cant is estimated using TrOCA method (refs doing the same with TrOCA in Sothern Ocean). » Missing words and references in the sentence ? Also typo: Southern Ocean.

C-06: Line 110: Authors used GLODAPv2 data (version GLODAPv2.2020). Are you selecting all the TA and DIC data or only samples when nutrients are all available (Nitrates, silicates and phosphates), i.e. samples without phosphates not included. Also, are you using samples with only TA data or with only DIC data ? Please clarify.

C-07: Line 130: Table 1: maybe specify in the caption that data used are in the SO south of 30°S.

C-08: Line 158: "…direct nitrate and pH measurements from Argo floats are excluded from this study." If I understand authors did not used pH and nitrates data from BGC-Argo floats and used only T, S and O2 from these floats (correct ?). Please clarify. Maybe a table (as Table 1 for GLODAP) would help to know the final selection of data from floats.

C-09: Line 168: typo CDT: …blue for Argo only with CTD,

C-10: Line 185: Figure 2: In the Box "Step 3", add O2 in the list of parameters as O2 is used in TrOCA.

C-11: Line 200: Authors used the TrOCA method to derive Cant concentrations that has been successfully used and/or compared in the SO (e.g. Lo Monaco et al, 2025; Vazquez-Rodriguez et al, 2009; Mahieu et al, 2020; Metzl et al, 2024). Authors refer to Lo Monaco et al (2025) but this is not the correct reference for TrOCA (see reference below).  Maybe recall that method is not adapted to estimate Cant in surface layer. How to you extrapolate the Cant concentrations in surface or indicate that climatology is limited to 100m ?

C-12: Line 204:"Finally, the Cant values are scaled to the reference year". It would be useful to specify the reference year, here 2013.

C-13: For Cant did you get any negative value ?

C-14: Line 207: I guess Lewis et al (2021) not listed in references.

C-15: Line 330:  "Under the full-parameter pathway, Cant exhibits a slight negative bias relative to shipboard derived values (Cant_ship_M)," Curiosity: Is there any change of the bias over time or is it evaluated for the reference year only ?

C-16: Figure 6b: The map is not clear. Is it for surface water, for deep layers ? Please clarify in the caption.

C-17: Line 364: Climatological distribution. Could you recall the period for the climatology presented in Figure 7 and 8; it is a mean for any data spanning 1972-2025 or for a reference year in 2013 ?

C-18: Line 375: when describing the fronts plotted in figure 7 (STF, SAF, PF…), add a reference (Orsi et al 1995, other ?).

C-19: Figures 7 and 8. I think a map is missing: "Abyssal water" listed in the captions.

C-20: Figures 7 and 8. Maybe specify in the caption that the color scales are different for each map.

C-21: Figure 7f,g show high Cant in the SW Atlantic (>20 µmol/kg ?). This signal is not resolved for the O2-Float-grid (Figure 9a). A comment on this ? Is TrOCA not adapted here or a problem with reconstruction ? (see also comment C-27 below)?

C-22: Figures 9a: Few data show very high Cant in the Pacific and Indian oceans (red points apparently > 25 µmol/kg). Are the data correct ? The maps 9a and 9c on top (shipboard) are small and it would be nice to show enlarged maps in Supp Mat.

C-23: Figures 9a and 9c: Could you recall in the caption the layer for the maps in deep layers (2000-4000 db ?)

C-24: Figures 9: the scales at the bottom is not clear (or not fully plotted)

C-25: Figures 9b: would it be possible to change the color for the lines (not clear for GLODAP: green in legend ?).

C-26: Line 417: "The TRACE-derived dataset (purple lines) consistently underestimates Cant relative to the TrOCA-derived values, particularly in intermediate waters." WHY ?

C-27: Line 426: "Notably, we identify a hotspot of high Cant concentrations in the southwestern Atlantic Ocean near the SAF and PF (Figure 9a), potentially linked to AABW outflow from the Weddell Sea". Intriguing signal (see comment C-21). If the selected depth for Figure 9a is 2000-4000db, would the signal linked to waters above the AABW (e.g. WSDW) or other origin (see for example Gruber, 1998; Ríos et al, 2012). Note also that Müller et al (2023) identified a relatively high increase of the Cant inventory in this region between 2004 and 2014 (for the layer 0-3000m). It would be interesting to show a section of Cant (Lat/depth) crossing this region in the Appendix. See for example figure R1 in this review. Would that anomaly in the SW Atlantic help to check the data or the methods ?

C-28: Line 453: "Although our approach may underestimate uncertainty due to potential representativity error, our dataset offers a significant improvement in both accuracy and spatial representativeness over previous gap-filling approaches". Maybe also recall that your product is extended to the bottom whereas previous analyses were mainly limited to the layer 0-3000m (e.g. Gruber et al, 2019).

C-29: Figure A6: For the abyssal water, the maps are not clear. Why there are so few data. Also change the code for the panels (a), (b, (c), (d) and (e). (not (f)) . As this layer is rather new compared to previous products I think this should be highlighted (move A6 to the main text ?). Are the front useful in these maps ?

C-30: Figures A8 and A9: Missing units for these maps.

;;;;;;; in references:

C-31: Line 607: Clement and Gruber (2018) not listed in the manuscript.

C-32: Line 656: Change the reference:
Lo Monaco C., C. Goyet, N. Metzl, A. Poisson and F. Touratier, 2005. Distribution and inventory of anthropogenic CO2 in the Southern Ocean : comparison of three data-based methods. Journal Geophys. Res. . 110, C09S02, doi:10.1029/2004JC002571.

;;;;;; Reference added in this review not listed in the Manuscript:

Gruber, N., 1998. Anthropogenic CO2 in the Atlantic Ocean. Global Biogeochem. Cycles, 12, 165–191.

Jiang, L.-Q., et al.,: Synthesis of data products for ocean carbonate chemistry, Earth Syst. Sci. Data Discuss. [preprint], https://doi.org/10.5194/essd-2025-255, in review, 2025.

Keppler, L., Landschützer, P., Gruber, N., Lauvset, S. K., and Stemmler, I.: Seasonal carbon dynamics in the near-global ocean, Global Biogeochemical Cycles, 34(12), e2020GB006571, https://doi.org/10.1029/2020GB006571, 2020.

Keppler, L., Landschützer, P., Lauvset, S. K., and Gruber, N.: Recent trends and variability in the oceanic storage of dissolved inorganic carbon, Global Biogeochemical Cycles, 37(5), https://doi.org/10.1029/2022gb007677, 2023.

Lauvset, S. K, R. M. Key, A. Olsen, S. van Heuven, A. Velo, X. Lin, C. Schirnick, A. Kozyr, T. Tanhua, M. Hoppema, S. Jutterström, R. Steinfeldt, E. Jeansson, M. Ishii, F. F. Pérez, T. Suzuki & S. Watelet, 2016. A new global interior ocean mapped climatology: the 1°x1° GLODAP version 2. Earth Syst. Sci. Data, 8, 325-340, doi:10.5194/essd-8-325-2016.

Lo Monaco C., C. Goyet, N. Metzl, A. Poisson and F. Touratier, 2005. Distribution and inventory of anthropogenic CO2 in the Southern Ocean : comparison of three data-based methods. Journal Geophys. Res. . 110, C09S02, doi:10.1029/2004JC002571.

Mahieu, L., Lo Monaco, C., Metzl, N., Fin, J., and Mignon, C.: Variability and stability of anthropogenic $CO_2$ in Antarctic Bottom Water observed in the Indian sector of the Southern Ocean, 1978–2018, Ocean Sci., 16, 1559–1576, https://doi.org/10.5194/os-16-1559-2020, 2020.

Metzl, N., et al: Anthropogenic CO2, air–sea CO2 fluxes, and acidification in the Southern Ocean: results from a time-series analysis at station OISO-KERFIX (51° S–68° E), Ocean Sci., 20, 725–758, https://doi.org/10.5194/os-20-725-2024, 2024.

Müller, J. D. , N. Gruber, B. Carter, R. Feely, M. Ishii, N. Lange, S. K. Lauvset, A. Murata, A. Olsen, F. F. Pérez, C. Sabine, T. Tanhua, R. Wanninkhof, D. Zhu, Decadal trends in the oceanic storage of anthropogenic carbon from 1994 to 2014. AGU Adv. 4, e2023AV000875 (2023).

Orsi, A. H., T. Whitworth III, and W. D. Nowlin Jr., On the meridional extent and fronts of the Antarctic Circumpolar Current, Deep Sea Res., Part I, 42, 641-673, 1995.

Pardo, P. C., et al,: Carbon uptake and biogeochemical change in the Southern Ocean, south of Tasmania. Biogeosciences, 14(22), 5217–5237. https://doi.org/10.5194/bg-14-5217-2017, 2017

Ríos, A. F., Velo, A., Pardo, P. C., Hoppema, M., and Pérez, F. F.: An update of anthropogenic CO2 storage rates in the western South Atlantic basin and the role of Antarctic Bottom Water, J. Mar. Syst., 94, 197–203, https://doi.org/10.1016/j.jmarsys.2011.11.023, 2012.

Rödenbeck, C., et al, 2015. Data-based estimates of the ocean carbon sink variability – First results of the Surface Ocean pCO2 Mapping intercomparison (SOCOM). Biogeosciences 12: 7251-7278. doi:10.5194/bg-12-7251-2015..

Vazquez-Rodriguez, M., et al, 2009. Anthropogenic Carbon Distributions in the Atlantic Ocean: data-based estimates from the Arctic to the Antarctic. Biogeosciences, 6, 439-451. https://doi.org/10.5194/bg-6-439-2009

;;;;;;;;;; Figure for review

Figure R1; Section of Cant (µmol/kg) along cruises in the South-Western Atlantic (GLODAP data in 2018-2019, Expocode: 29HE20190406 and 74JC20181103). In deep layers, below 1500m, there is no concentration higher than 20 µmol/kg as suggested in Figured 7 f,g.

[Figure]

;;;;;;;;;;;;;;;;;; end review

---

## Author Response (AR1)

Dear Editor, Dear Reviewers,

We sincerely appreciate your careful reviews and insightful comments, which have greatly helped us improve the clarity, quality, and impact of this study. We have taken all the comments and suggestions into account in a thorough revision of our manuscript. We are happy to report that our key messages remain unchanged and are strengthened. Here we briefly summarize the most significant modifications as follows:

The reviewers offered four main suggestions for improvement: (1) explaining in detail the signal of the gridded $C_{ant}$ product in the southwestern Atlantic, (2) improving the spatial distribution and other figure illustrations, (3) clarifying the data processing workflow (including whether sea-ice effects need to be considered), gridding methodology, and uncertainty estimation, and (4) optimizing dataset publication, including the output variables and data presentation format.

(1) Per reviewer #1 suggestion, we used BGC-Argo floats equipped with $O_2$ sensors to compare the hydrography-only and full-parameter pathways, thereby better quantifying the differences between the two. We found that the signal in the southwestern Atlantic from the Non-$O_2$-Float Grid likely results from limited model training and recommend using the $O_2$-Float Grid for this region.

(2) Per reviewer #1 and #2 comment, we now use depth in meters to be consistent with most other ocean carbon data products, and have moved the spatial distribution and discussion of the abyssal layer from the Appendix to the main text. The corresponding figures were revised to improve readability and make the spatial patterns and analyses more intuitive.

(3) Per reviewer #1 and #2 comments, we clarified the treatment of negative $C_{ant}$ values, and provided a more detailed explanation of the gridding procedure, uncertainty estimation, and quality control process. Additionally, preliminary tests show that the inclusion of sea-ice effects remains limited at this stage.

(4) Per reviewer #2 suggestion, we expanded our product to include additional carbonate system variables—carbonate ion concentration, calcite saturation state, and the Revelle Factor. We also refined the vertical layer structure and transferred the dataset to NOAA's Ocean Carbon and Acidification Data System (OCADS) at NCEI (in submission).

All reviewer comments have been addressed point-by-point in the attached response document (author replies are marked in blue).

The revised manuscript meets the formatting requirement of *Earth System Science Data*. It now contains 6893 words in the main text (10 figures; 66 references), two Appendix (1634 words, 3 tables, 14 figures).

We believe that these changes, together with other suggested edits, have substantially improved the quality and impact of the manuscript.

Sincerely yours,

Di Qi on behalf of all authors

**Response to Referee #1 (R1):**

In this paper authors present a new dataset of the carbonates system, called SOCOML (Southern Ocean CO2 Machine Learning), derived from shipboard (GLODAP) and float (Argo) data in the Southern Ocean (here south of 30°S). This is a climatological product that includes aragonite saturation state ($\Omega$ar) and anthropogenic CO2 concentrations (Cant) for a reference year (2013). Such climatology (for DIC and TA) has been previously developed, based on shipboard data (e.g., Lauvset et al, 2016; Keppler et al 2020, 2023). Like for surface pCO2 and air-sea CO2 fluxes products (e.g. SOCOM project, Rödenbeck et al, 2015) successfully used to constraint the estimate of the global carbon budget (Friedlingstein et al, 2025) it is important that different climatology for the ocean interior are available for the community (see also a list of such products in Jiang et al, 2025).

After a clear introduction, authors describe the methodology and uncertainties. Three gridded products are offered (with or without O2) at 1x1 degree resolution and 84 levels in the water column down to the abyssal domain. This is especially important in the Southern Ocean where AABW are formed and contain significant Cant concentrations (e.g. Rios et al, 2012; Pardo et al, 2017; Mahieu et al, 2020; Zhang et al 2023). This is also challenging as previous analyses explored the changes of Cant inventories over 0-3000m (Gruber et al, 2019; Müller et al, 2023). Here the product extend to the bottom.

The reconstructed fields for AT, DIC, O2 and nutrients are used to calculate $\Omega$ar and Cant concentrations (here derived from TrOCA method). I suspect this new data-sets could be easily used to estimate Cant using other methods. The results (climatology of TA, DIC, and Cant concentrations) are also important to validate OBGM and ESM models that suffer to reproduce seasonal to multi-decadal DIC and Cant variability. The results will be also probably useful for paleo-oceanography studies (e.g. using pre-industrial DIC profiles deduced from the data presented in this work when calculate DIC-Cant).

Interestingly, through a comparison of different results, authors indicate that the bias in DIC and Cant depend on the dataset used for the training, highlighting the need of maintaining regular update of the dataset such as GLODAP. This is especially true in region where data are relatively sparse such as near coastal Antarctica and in the seasonal ice zone during austral winter. By the way, I was wondering how the sea-ice is taken into account in the method (or if it should be taken into account).

Thank you for your positive and constructive summary of our work. The comments are detailed and helpful in preparing the revised manuscript.

Regarding the comment on whether sea ice should be considered: it is indeed an important factor influencing the carbonate system in the Southern Ocean through its modulation of air–sea $CO_2$ exchange and brine-driven processes. To examine whether sea-ice information should be explicitly included as a predictor in the models, we conducted an exploratory test using the monthly sea-ice concentration (SIC) and sea-ice thickness (SIT) product of Fons et al. (2023) covering 2010–2021. SIC represents

the fractional areal coverage of sea ice, whereas SIT describes its vertical depth. Figure R1 below illustrates that sea ice generally persists along the Antarctic coastal margin up to about 70° S, expanding northward during winter, particularly in the Atlantic Sector where it can reach as far as ~55° S.

Machine-learning framework requires one-to-one matched predictors and targets, we therefore collocated SIC and SIT data with the GLODAPv2.2022 for model training. The seasonal ice zone (SIZ; Fig. a) exhibits SIC and SIT variability, but most cruises were conducted during austral summer (mainly in January). South of 55° S, 77 % of the GLODAP measurements have SIC = 0 and 91 % have missing (NaN) SIT values. This indicates that, although sea ice is physically important, it cannot be meaningfully included as an input variable for most samples, nor can machine learning effectively capture transitions from fully ice-covered (SIC ≈ 100 %) to open-ocean (SIC ≈ 0 %) conditions.

Nevertheless, we conducted preliminary machine-learning experiments to evaluate this possibility. In previous work (Carter et al., 2021), separate neural networks were trained for the Atlantic–Mediterranean–Arctic and Indo–Pacific–Southern sectors to ensure regional representativeness. Within this domain, however, 80–90 % of measured values were associated with missing SIC/SIT data, and even when available, the variability was small—rendering the inclusion of SIC/SIT as predictors ineffective.

To maximize the available samples, we applied monthly SIT fields at each location (month-matched), which yielded a collocated dataset of 47,913 pairs within the SIZ. Among these, approximately 51 % and 32 % contained measured DIC and TA, respectively. We then trained prototype ESPER_NN models using 85% of the collocated dataset for training and 15% for validation, with temperature, salinity, oxygen, and SIC/SIT as predictors and DIC as the target variable. The models did not achieve satisfactory performance: the $R^2$ remained low regardless of whether sea-ice information was included. This weak performance likely results from the limited measurements south of 60° S, which is insufficient to support robust regional training. Moreover, since the predictors and targets are three-dimensional (longitude, latitude, pressure/depth), whereas SIC/SIT is two-dimensional (longitude, latitude), only about 1,880 unique SIT values provided independent information after accounting for repeated depths.

Physically, sea ice primarily affects the surface and upper mixed layer by air–sea exchange and altering brine processes. Its influence on deeper waters is indirect and occurs mainly through processes associated with the formation of Antarctic Bottom Water (AABW). Based on the spatial distribution of sea-ice cover, this influence is expected to be exerted primarily through AABW formation regions. However, because our product targets the full water column of entire Southern Ocean, the overall impact of sea-ice processes on the deeper ocean interior remains limited. Given the sparse

sampling in ice-covered region and the limited vertical representativeness of sea-ice data, we therefore did not include it as an explicit predictor.

Nevertheless, incorporating sea-ice information is highly relevant for studies focusing on the Antarctic coastal ocean and for reconstructions of surface-ocean properties such as $pCO_2$ and chlorophyll, which differ in focus from our dataset that targets the broader Southern Ocean interior.

[Figure]

**Figure R1**. Distribution of sea–ice extent and concentration in the Southern Ocean. (a) Seasonal iced zones derived from sea-ice thickness climatology. Red dots indicate locations of GLODAPv2 observations used in this study. (b) Annual mean sea-ice concentration averaged over 2010–2021.

Fons, S., Kurtz, N., and Bagnardi, M.: A decade-plus of Antarctic sea ice thickness and volume estimates from CryoSat-2 using a physical model and waveform fitting, The Cryosphere, 17, 2487-2508, 10.5194/tc-17-2487-2023, 2023.

The manuscript is clear, tables and figures adapted. The dataset will be certainly useful for many studies including validation of models. I recommend publication after some clarifications. Below are listed specific and minor comments.

We sincerely thank the reviewer for the positive and encouraging evaluation. We have carefully addressed all specific and minor comments point by point. The line numbers mentioned in our responses refer to the revised version with *Track Changes* enabled for review.

;;;;;;;;;; Specific comments:

C-01: Title: "High-resolution spatiotemporal fields of Southern Ocean interior carbonate system parameters integrated from float- and ship-based observations". The products are at spatial scale of 1 x 1 degree (not really high-resolution) for a reference year and include anthropogenic CO2 (Cant) so I would suggest change the title: "Climatological fields of Southern Ocean interior carbonate system parameters and anthropogenic CO2 integrated from float- and ship-based observations".

Thank you for your valuable suggestions. Following your recommendation, and in line with Reviewer 2's comment, we have revised the title to:

"*Climatological fields of Southern Ocean interior carbonate system parameters and anthropogenic $CO_2$ reconstructed and integrated from float- and ship-based observations*".

C-02: Line 68: « while the Global Ocean Data Analysis Project version 2 (GLODAPv2, Olsen et al., 2016) offers quality-controlled data product from the surface into the ocean interior includes TA and DIC." It also offered climatology of TA, DIC in the interior ocean (Lauvset et al, 2016).

Thank you for pointing this out and we have now updated the description to the sentence (Line 68).

"…, while the Global Ocean Data Analysis Project version 2 (GLODAPv2, Olsen et al., 2016) offers quality-controlled data as well as climatological products (Lauvset et al, 2016) from the surface into the ocean interior, including TA and DIC."

C-03: Line 70: Maybe also refer here to the last RECCAP-2 story for the SO (Hauck et al, 2023).

We have now added the reference of Hauck et al, 2023 (Line 71).

C-04: Line 99: "TA, DIC, pH (total scale), nitrate (NO3), phosphate (PO4) and silicate (SiO4) are obtained through neural networks". Add also O2 in this list as this is used for Cant calculation ?

We have now included $O_2$ in the list and clarified that it is only estimated when direct measurements are not available. The revised sentence now reads (Line 102):

"TA, DIC, pH (total scale), nitrate ($NO_3$), phosphate ($PO_4$), silicate ($SiO_4$), and $O_2$ (when direct measurements are unavailable) are obtained through neural networks,…"

C-05: Line 100-101: « And Cant is estimated using TrOCA method (refs doing the same with TrOCA in Sothern Ocean). » Missing words and references in the sentence ? Also typo: Southern Ocean.

We have added the missing reference (Zhang et al., 2023 and Metzl et al., 2024) and corrected the typo.

C-06: Line 110: Authors used GLODAPv2 data (version GLODAPv2.2020). Are you selecting all the TA and DIC data or only samples when nutrients are all available (Nitrates, silicates and phosphates), i.e. samples without phosphates not included. Also, are you using samples with only TA data or with only DIC data ? Please clarify.

We apologize for the confusion caused by our previous wording. In the revised manuscript, we specify that the data version used is GLODAPv2.2023, to incorporate as many ship-based observations as possible.

It should be clarified that the GLODAP versions used to train the reconstruction models for TA, DIC, pH, and nutrients differ. To enable consistent comparisons in the Southern Ocean, and in particular to evaluate potential effects of temporal trends on model performance, we additionally employed GLODAPv2.2020. The new records included in GLODAPv2.2023 compared to GLODAPv2.2020 were not used in any model training, and thus served as an independent assessment dataset (Table 1) to evaluate model accuracy. In contrast, the Total data listed in Table 1 were used to generate the gridded climatological fields. This explanation also addresses the concern raised in C-15, which we discuss further below.

Regarding the reviewer's question on the use of TA and DIC data: after quality control, we retained TA and DIC samples only when concurrent nutrient measurements (nitrate, phosphate, and silicate) were available, following Carter et al. (2021a). To avoid ambiguity, the text has been revised (Lines 122) as follows:

"*Importantly, quality control is performed independently for each variable. Subsequently, TA and DIC measurements are retained only when nutrient observations are available, following Carter et al. (2021a).*"

C-07: Line 130: Table 1: maybe specify in the caption that data used are in the SO south of 30°S.

Agreed – added in the caption (Line 135).

C-08: Line 158: "…direct nitrate and pH measurements from Argo floats are excluded from this study." If I understand authors did not used pH and nitrates data from BGC-Argo floats and used only T, S and O2 from these floats (correct ?). Please clarify. Maybe a table (as Table 1 for GLODAP) would help to know the final selection of data from floats.

Your understanding is correct. To clarify, we have revised the sentence as:

"…nitrate and pH measurements from BGC-Argo floats are not used in this study; only temperature, salinity, and $O_2$ are employed." (Line 164)

In addition, we have added the statistics of Argo float profiles to Table 1 and revised the caption accordingly, as shown below (Line 135):

**Table 1.** Numbers of shipboard GLODAPv2 measurements and Argo float profiles for each variable in the Southern Ocean used in this study. The assessment dataset of GLODAPv2 data product used for mode-performance comparisons contains cruises added after the GLODAPv2.2020 release—specifically, those with cruise identifiers ≥ 2107.

|  | Variables | Oxygen (Y/N) | Assessment dataset | Total |
|---|---|---|---|---|
| Shipboard | TA | N | 15,059 | 103,140 |
| GLODAPv2 | | Y | 14,954 | 101,870 |
| measurements | DIC | N | 15,376 | 129,799 |

| | | | | |
|---|---|---|---|---|
| | | Y | 15,270 | 126,312 |
| | pH | N | 14,996 | 56,597 |
| | | Y | 14,894 | 56,411 |
| | NO$_3$ | N | 18,796 | 232,771 |
| | | Y | 18,575 | 226,665 |
| | PO$_4$ | N | 18,352 | 224,262 |
| | | Y | 18,086 | 218,876 |
| | SiO$_4$ | N | 19,011 | 242,736 |
| | | Y | 18,745 | 234,807 |
| Argo float | Temp, Sal | - | - | 647,650 |
| profiles | Temp, Sal, Oxy | - | - | 73,296 |

C-09: Line 168: typo CDT: …blue for Argo only with CTD,

Corrected (Line 172).

C-10: Line 185: Figure 2: In the Box "Step 3", add O2 in the list of parameters as O2 is used in TrOCA.

We have updated the Figure 2 (Line 190):

[Figure]

C-11: Line 200: Authors used the TrOCA method to derive Cant concentrations that has been successfully used and/or compared in the SO (e.g. Lo Monaco et al, 2005; Vazquez-Rodriguez et al, 2009; Mahieu et al, 2020; Metzl et al, 2024). Authors refer to Lo Monaco et al (2005) but this is not the correct reference for TrOCA (see reference

below). Maybe recall that method is not adapted to estimate Cant in surface layer. How to you extrapolate the Cant concentrations in surface or indicate that climatology is limited to 100m ?

1) The reference to Lo Monaco et al, 2005 has been corrected, and the additional references suggested have been incorporated. The revised sentence now reads:

"*For these reasons, this study employs the TrOCA method, which is relatively straightforward, extensively utilized in Southern Ocean studies (Metzl et al., 2024; Zhang et al., 2023), and has been demonstrated to be reliable through comparative analyses (Lo Monaco et al., 2005; Mahieu et al., 2020; Vázquez-Rodríguez; Zhang et al., 2023).*" (Line 205)

2) With respect to the concern on surface values, we clarify that the calculation of $C_{ant}$ is limited to waters below the euphotic layer (about 100 m). This restriction has been explicitly added in Section 3.2 and in the dataset metadata. The clarification reads:

"It should be noted that the TrOCA approach is limited to waters below the euphotic layer, therefore the $C_{ant}$ estimates above 100 m are excluded from this dataset." (Line 212)

C-12: Line 204:"Finally, the Cant values are scaled to the reference year". It would be useful to specify the reference year, here 2013.

We appreciate the suggestion. We have now specified it here. (Line 210)

C-13: For Cant did you get any negative value?

We acknowledge that this point was not clearly explained earlier. A small number of negative $C_{ant}$ values were identified, which account for only 0.003 % of all samples and appear in about 1 % of the total profiles. An example of an Argo float showing negative values is displayed in the figure below. These negative estimates most likely result from the propagation of methodological and observational uncertainties. It is worth noting that the mean of all negative values is $-1.08$ µmol kg$^{-1}$, and their deviation from zero generally falls within the estimated uncertainty range of $C_{ant}$ of approximately ±6 µmol/kg. Therefore, although negative $C_{ant}$ values are physically unrealistic, they are mathematically reasonable within the context of uncertainty propagation. To ensure consistency in the mathematical treatment of error propagation, we followed the approach of Gruber et al. (1998) and retained these negative $C_{ant}$ values in the averaging process to avoid introducing artificial biases into the gridded fields. To clarify this point, we have added the following sentence to the main text:

"Following Gruber et al., 1998, negative $C_{ant}$ value is preserved in the averaging process." (Line 242)

[Figure]

**Figure R2**. Vertical distribution of anthropogenic carbon ($C_{ant}$) estimated using Argo float (WMO: 6902995). Each grey dot represents an individual estimate and the red line at zero indicates the reference for distinguishing positive and negative $C_{ant}$ values.

C-14: Line 207: I guess Lewis et al (2021) not listed in references.

Thank you for noticing this oversight. The reference has been corrected and listed: it should be *Sharp et al., 2023* instead of *Lewis et al., 2021*. (Line 214)

Sharp, J. D., Pierrot, D., Humphreys, M. P., Epitalon, J.-M., Orr, J. C., Lewis, E. R., and Wallace, D. W. R.: CO2SYSv3 for MATLAB (Version v3.2.1), Zenodo [code], https://doi.org/10.5281/zenodo.3950562, 2023.

C-15: Line 330: "Under the full-parameter pathway, Cant exhibits a slight negative bias relative to shipboard derived values (Cant_ship_M)," Curiosity: Is there any change of the bias over time or is it evaluated for the reference year only?

This is an important question. When evaluating the bias of float-derived values relative to shipboard-derived values (Cant_ship_M), we initially did not apply normalization. We have now recalculated the normalized bias, and the updated results are shown in the revised Figure 4 (Line 365).

[Figure]

In addition, we examined whether machine-learning reconstructions can capture temporal trends in $C_{ant}$. A scatterplot of bias versus year (see below) shows no evident trend. It is nevertheless likely that biases may emerge in future years due to the exponential increase in atmospheric $CO_2$, particularly in the Southern Ocean where

$C_{ant}$ penetration is more pronounced. However, empirical approaches such as machine learning risk projecting natural variability into apparent long-term trends (Carter et al., 2021), as they are not designed to resolve non–steady-state variations in $C_{ant}$.

In practice, most machine-learning models are constructed under the assumption that the ocean operates in a steady state. For example, Sauzède et al. (2017) and Bittig et al. (2018) incorporated sampling date information directly into the input parameters, whereas Carter et al. (2021) developed the ESPER model by adjusting DIC and pH values to a reference year before training and then re-adjusting them to the desired year at the output stage. Both approaches rely on the assumption of short-term oceanic steady state within the sampling period of the training data. This methodological constraint limits the use of machine-learning reconstructions to assess trends in the bias of DIC and $C_{ant}$.

For these reasons, in this study we chose to present climatological distributions of carbonate system parameters and nutrients rather than focus on temporal trends. Nonetheless, we acknowledge that developing approaches for machine learning to better capture long-term trends and potentially seasonal variability remains an important direction for future work.

[Figure]

**Figure R3**. Temporal distribution of biases in $C_{ant}$ estimates.

C-16: Figure 6b: The map is not clear. Is it for surface water, for deep layers? Please clarify in the caption.

Thank you for this observation. The comparisons in Figure 6b are restricted to the 1,400-2,100 dbar depth range. We have clarified this in both the caption (Line 374) and the manuscript text (Line 353).

C-17: Line 364: Climatological distribution. Could you recall the period for the climatology presented in Figure 7 and 8; it is a mean for any data spanning 1972-2025 or for a reference year in 2013 ?

Figure 7 and 8 illustrate the spatial distribution of the Float Grid, as noted in Line 372. The climatology represents a mean of Argo profile data spanning 2001-2024 for TA,

DIC, pH, Ω$_{ar}$, and nutrients, while C$_{ant}$ is expressed as a mean for a reference year 2013. We have clarified this in the captions of Figure 7, 8, A4, A5, and A6.

C-18: Line 375: when describing the fronts plotted in figure 7 (STF, SAF, PF…), add a reference (Orsi et al 1995, other ?).

A reference to Orsi et al. (1995) has been added when describing the fronts plotted in Figures to provide appropriate citation.

C-19: Figures 7 and 8. I think a map is missing: "Abyssal water" listed in the captions.

Sorry for our incaution. The caption has been revised, and the abyssal-layer distributions have now been incorporated into the main text, following the related suggestion in Comment C-29.

C-20: Figures 7 and 8. Maybe specify in the caption that the color scales are different for each map.

We have revised the caption by adding the statement "Note that the color scales differ among the individual maps" at the end.

C-21: Figure 7f,g show high Cant in the SW Atlantic (>20 µmol/kg ?). This signal is not resolved for the O2-Float-grid (Figure 9a). A comment on this ? Is TrOCA not adapted here or a problem with reconstruction? (see also comment C-27 below)?

Figure R1; Section of Cant (µmol/kg) along cruises in the South-Western Atlantic (GLODAP data in 2018-2019, Expocode: 29HE20190406 and 74JC20181103). In deep layers, below 1500m, there is no concentration higher than 20 µmol/kg as suggested in Figured 7 f,g.

[Figure]

Thank you for prompting us to reconsider this point carefully. Following your comment (see also C-27), we conducted a detailed re-examination of the parameter distributions in the southwestern Atlantic (SW Atlantic; 60°W–30°W, 30°S–60°S).

(1) High C$_{ant}$ concentrations (>20 µmol/kg)

We first inspected the GLODAP data in this region. As you noted, for the two cruises (Expocode: 29HE20190406 and 74JC20181103), the vertical profiles of Cant

calculated using the TrOCA method (Figure R4a) show no concentrations exceeding 20 µmol kg⁻¹ in the deep ocean, consistent with your observation. It should be noted that the DIC data of cruise 29HE20190406 south of 30°S have quality flags = 0; therefore, these data were excluded from our DIC and $C_{ant}$ analysis. Figure R4b presents all TrOCA-derived $C_{ant}$ data from GLODAP within the SW Atlantic, revealing a few isolated deep-ocean values above 20 µmol kg⁻¹. Moreover, Zhang et al. (2023) also reported scattered > 20 µmol kg⁻¹ values in the deep waters of the Weddell Sea and the adjacent SW Atlantic based on three independent methods for estimating $C_{ant}$ (Figure R5), suggesting that the TrOCA approach remains applicable in this region.

[Figure]

**Figure R4**. Vertical distribution of $C_{ant}$ in the southwest Atlantic from GLODAPv2.2023.

[Figure]

**Figure R5**. Comparison of $C_{ant}$ estimates in the SW Atlantic Ocean near Weddell Sea (Zhang et al., 2023).

(2) Difference between O2-Float Grid and non-O2-Float Grid

Temperature and salinity profiles from Core and BGC Argo floats are generally consistent, and the distributions of dissolved oxygen measured by BGC Argo floats agree well with those reconstructed from Core Argo data (Figure B3), confirming that the $O_2$ reconstruction in this region is reliable. However, the profiles of TA, DIC, and consequently $C_{ant}$ and $\Omega_{ar}$ exhibit noticeable differences (Figures B4), arising from the

inherent limitations of the machine-learning model that lead to distinct reconstruction pathways.

In Section 4.2, we analyzed in detail how the two reconstruction pathways influence $C_{ant}$ and $\Omega_{ar}$ based on shipboard measurements. We further assessed these pathway differences using float data located near the shipboard sites, focusing on a restricted depth range (1,400–2,100 dbar) and comparable seawater property conditions. From this joint analysis of GLODAP and float data, we found that the differences in float-based $C_{ant}$ and $\Omega_{ar}$ between the two pathways are within ±10 µmol kg$^{-1}$ and ±0.075 unit, respectively.

However, in regions with sparse training data—such as the SW Atlantic—our gridded product exhibits noticeably larger discrepancies between the $O_2$-Float and non-$O_2$-Float Grid products. This reflects a common limitation of neural-network and other regression-based methods. To quantify this, we used BGC-Argo floats carrying $O_2$ sensors to compare the hydrography-only pathway (T, S) against the full-parameter pathway (T, S, $O_2$). For each 1° × 1° bin, we calculated the mean vertical difference in $C_{ant}$ and $\Omega_{ar}$ between the two pathways, and the resulting distribution is shown in Figure B2. As the full-parameter pathway includes observed $O_2$ and thus yields more constrained results, we consider its reconstructed $C_{ant}$ and $\Omega_{ar}$ to be more accurate. The comparison reveals a clear overestimation of $C_{ant}$ and localized underestimation of $\Omega_{ar}$ in the SW Atlantic under the *hydrography-only* pathway. Similar but weaker differences are observed near the Ross Sea (overestimated $C_{ant}$, underestimated $\Omega_{ar}$) and along the Pacific coast of South America (underestimated $C_{ant}$, overestimated $\Omega_{ar}$), although their spatial patterns are much less pronounced. The pronounced differences in the SW Atlantic arises from the limited number of shipboard training samples in this region—only about 1.3 % of the quality-controlled GLODAPv2.2023 measurements in the Southern Ocean contain both TA and DIC measurements. The performance of the machine-learning reconstructions and our gridded dataset is expected to improve as more such high-quality data become available in this area. In the SW Atlantic, $C_{ant}$ exhibits a more stratified vertical structure rather than the deep-penetrating signal captured by the non-$O_2$-Float grid. Therefore, for studies focused on specific regions such as the SW Atlantic, we recommend using the $O_2$-Float Grid provided in this study.

(3)Revision in the main text

Accordingly, we have revised the relevant section in the manuscript to clarify this issue and have added a brief discussion on the regional limitation of the reconstruction pathways in the SW Atlantic in Appendix B.

a. **Appendix B2. Differences between reconstruction pathways for float-based products**

[revised manuscript text omitted]

C-22: Figures 9a: Few data show very high Cant in the Pacific and Indian oceans (red points apparently > 25 µmol/kg). Are the data correct ? The maps 9a and 9c on top (shipboard) are small and it would be nice to show enlarged maps in Supp Mat.

We appreciate this careful observation. Figure 9a is intended to present the climatological distribution of $C_{ant}$ within the 1400–2000 dbar layer (intermediate layer, consistent with Figure 7f). However, we mistakenly indicated a deeper layer in the original caption. This has now been corrected. In addition, enlarged versions of the shipboard-based maps have been added to Figure A9 after consideration of the reviewer's suggestion.

[Figure]

**Figure A9.** Averaged climatological distribution of $C_{ant}$ (a-d) and $\Omega_{ar}$ (e-h) in oceanic sectors and three layers: subsurface layer (100 to 300 m), intermediate layer (1400 to 2000 m), deep layer (2000 to 4000 m), and abyssal layer (4000 to 5600 m). The climatology is derived from GLODAPv2.2023, while $C_{ant}$ is scaled to the reference year 2013. The thin black lines show, from north to south, the Subtropical Front (STF), the Subantarctic Front (SAF), the Polar Front (PF), and the Southern Antarctic Circumpolar Current Front (SACCF) (Oris et al., 1995). Note that the color scales differ among the individual maps.

C-23: Figures 9a and 9c: Could you recall in the caption the layer for the maps in deep layers (2000- 4000 db ?)

The caption has been revised to specify the pressure range of layers.

C-24: Figures 9: the scales at the bottom is not clear (or not fully plotted)

We have revise the color scale of $\Omega_{ar}$ and updated in Figure 9 and corresponding Figure 8g.

C-25: Figures 9b: would it be possible to change the color for the lines (not clear for GLODAP: green in legend ?).

We acknowledge the inconsistency in the previous color scheme for GLODAP between the legend and the subsurface panels. We have corrected this issue and replaced the original color with a clearer one to improve visual distinction. Together with the revisions made in response to C-23 and C-24, the figure has been updated accordingly as shown below (Line 470).

[Figure]

**Figure 10.** Panels (a) and (c) show the climatological distribution of $C_{ant}$ and $\Omega_{ar}$ in the intermediate layers (1400-2000 m), respectively. Panel (b) and (d) present latitudinal distributions of $C_{ant}$ and $\Omega_{ar}$ averaged over four depth layers: subsurface layer (100 to 300 m), intermediate layer (1400 to 2000 m), deep layer (2000 to 4000 m), and abyssal layer (4000 to 5600 m). Black, red, blue, green and purple symbols and lines represent Non-$O_2$-Float Grid, Float Grid, All-Data Grid, GLODAP-derived data, and TraceV1-derived data, respectively.

C-26: Line 417: "The TRACE-derived dataset (purple lines) consistently underestimates Cant relative to the TrOCA-derived values, particularly in intermediate waters." WHY ?

We acknowledge that our original wording was misleading. The intention was not to suggest that TRACE underestimates or TrOCA overestimates $C_{ant}$, but simply to indicate that the TRACE-derived dataset yields lower $C_{ant}$ concentrations than the TrOCA-derived values.

According to Carter et al. (2025), TRACEv1 is essentially an observation-tuned, model-based product in the deep ocean but shows limitations in regions where relatively young waters mix with older deep waters in substantial amounts (e.g., Antarctic Intermediate Water). In such regions, the one-dimensional idealized "pipe model" age distribution used in TRACEv1 is often inadequate, and the simple inverse Gaussian (IG) formulation is limited when reconstructing modelled $C_{ant}$.

In addition, TRACEv1 tends to exhibit larger reconstruction errors in regions with sparse training data. Because transient tracer measurements that include all three tracers (CFC-11, CFC-12, and SF$_6$) remain relatively rare—representing only about 5 %

globally of the GLODAPv2.2023 dataset. Our analysis shows that in the Southern Ocean, only 37,258 transient tracer samples are available, with merely ~10 % of them located in intermediate waters. Given the complex water-mass structure and strong upwelling in intermediate layers in 30° S-60° S, the scarcity of tracer observations likely contributes to discrepancies between the model-based TRACEv1 product and our observation-based TrOCA estimates.

We leave this issue for future work. With the continued expansion of seawater property and transient tracer measurements in the Southern Ocean, it will become possible to better evaluate the consistency among different anthropogenic carbon estimation methods and improve both empirical calculations and model simulations.

The corresponding section in the manuscript has been revised as follows:

"*Additionally, we apply the TRACE method (Carter et al., 2025) to estimate $C_{ant}$, generating a gridded dataset as described in Section 3.3, which serves as an additional comparison (Figure 9b). TRACEv1 adopts a hybrid conceptual framework, with surface-ocean estimates that are observation-based, whereas deep-ocean fields are model-based and tuned against observations.*" (Line 439)

"*The TRACE-derived dataset (purple lines) yields lower $C_{ant}$ concentration than the TrOCA-derived values, particularly in intermediate waters. This difference may partly arise because the model-based approach is difficult to constrain accurately in regions with sparse transient tracer observations and complex vertical structures associated with Southern Ocean upwelling (as suggested by Carter et al., 2025), leading to deviations from the observation-based TrOCA estimates.*" (Line.445)

C-27: Line 426: "Notably, we identify a hotspot of high Cant concentrations in the southwestern Atlantic Ocean near the SAF and PF (Figure 9a), potentially linked to AABW outflow from the Weddell Sea". Intriguing signal (see comment C-21). If the selected depth for Figure 9a is 2000-4000db, would the signal linked to waters above the AABW (e.g. WSDW) or other origin (see for example Gruber, 1998; Ríos et al, 2012). Note also that Müller et al (2023) identified a relatively high increase of the Cant inventory in this region between 2004 and 2014 (for the layer 0-3000m). It would be interesting to show a section of Cant (Lat/depth) crossing this region in the Appendix. See for example figure R1 in this review. Would that anomaly in the SW Atlantic help to check the data or the methods ?

;;;;;;;;;; Figure for review

Refer to C21.

C-28: Line 453: "Although our approach may underestimate uncertainty due to potential representativity error, our dataset offers a significant improvement in both accuracy and spatial representativeness over previous gap-filling approaches". Maybe also recall that your product is extended to the bottom whereas previous analyses were mainly limited to the layer 0-3000m (e.g. Gruber et al, 2019).

Thank you for this insightful suggestion. We have now revised the sentence in Line 479 accordingly and incorporated a brief reference to this point in the Introduction.

In the *Introduction*, we now introduce this aspect as a motivating point:

"In this study, we leverage the ESPER_NN model, integrating the high-accuracy GLODAPv2 database with profiling measurements of fine spatiotemporal resolution, to generate a comprehensive carbonate system dataset throughout the interior Southern Ocean that extends from the surface to the deep ocean (5,600 m)." (Line 99)

C-29: Figure A6: For the abyssal water, the maps are not clear. Why there are so few data. Also change the code for the panels (a), (b, (c), (d) and (e). (not (f)) . As this layer is rather new compared to previous products I think this should be highlighted (move A6 to the main text ?). Are the front useful in these maps ?

We acknowledge this valuable suggestion. To more clearly represent the observational coverage, our gridding procedure follows a straightforward bin-averaging approach without applying any horizontal interpolation or objective mapping. The scarcity of data in the abyssal layer of Float Grid mainly reflects the limited number of Deep Argo observations currently available at such depths. To better align with our study, which integrates both float- and ship-based observations, we have added the All-Data Grid distributions for the abyssal layer in the main text, providing the most comprehensive observation-based representation below 4,000 m. The frontal lines in the original abyssal-layer maps were included solely for consistency with the maps of other layers and did not serve additional interpretative purposes.

In addition, we have revised Section 4.3 to include the following description for the abyssal layer:

"Figure 9 presents the distributions in the abyssal layer. Because observations below 4,000 m are extremely scarce, the All-Data Grid is used to provide the most comprehensive depiction of deep-ocean conditions." (Line 384)

"In the deep and abyssal layer (2000-5600 m, Figure 9c), the spatial patterns of DIC and $C_{ant}$ remain unchanged, and their vertical profiles flatten. Both DIC and $C_{ant}$ exhibit relatively high concentrations in the eastern Antarctic region, where Antarctic Bottom Waters (AABW) forms (Morrison et al., 2020). This enrichment is consistent with AABW-driven transport of anthropogenic carbon into the deep ocean." (Line 404)

**Figure 9.** Averaged climatological distribution of DIC (a), $C_{ant}$ (b), pH (c), and $\Omega_{ar}$ (d) in the abyssal layer (4000 to 5600 m), The climatology is based on the All-Data Grid combined float- and ship-based observations. Note that the color scales differ among the individual maps.

C-30: Figures A8 and A9: Missing units for these maps.

The missing units have been added to the maps.

;;;;;;; in references:

C-31: Line 607: Clement and Gruber (2018) not listed in the manuscript.

We have removed the reference.

C-32: Line 656: Change the reference: Lo Monaco C., C. Goyet, N. Metzl, A. Poisson and F. Touratier, 2005. Distribution and inventory of anthropogenic CO2 in the Southern Ocean : comparison of three data-based methods. Journal Geophys. Res. . 110, C09S02, doi:10.1029/2004JC002571.

We have replaced the previous reference (Hauck et al., 2020) with Lo Monaco et al., 2005 (Line 756).

;;;;;;; Reference added in this review not listed in the Manuscript:

Gruber, N., 1998. Anthropogenic CO2 in the Atlantic Ocean. Global Biogeochem. Cycles, 12, 165–191.

Jiang, L.-Q., et al.,: Synthesis of data products for ocean carbonate chemistry, Earth Syst. Sci. Data Discuss. [preprint], https://doi.org/10.5194/essd-2025-255, in review, 2025.

Keppler, L., Landschützer, P., Gruber, N., Lauvset, S. K., and Stemmler, I.: Seasonal carbon dynamics in the near-global ocean, Global Biogeochemical Cycles, 34(12), e2020GB006571, https://doi.org/10.1029/2020GB006571, 2020.

Keppler, L., Landschützer, P., Lauvset, S. K., and Gruber, N.: Recent trends and variability in the oceanic storage of dissolved inorganic carbon, Global Biogeochemical Cycles, 37(5), https://doi.org/10.1029/2022gb007677, 2023.

Lauvset, S. K, R. M. Key, A. Olsen, S. van Heuven, A. Velo, X. Lin, C. Schirnick, A. Kozyr, T. Tanhua, M. Hoppema, S. Jutterström, R. Steinfeldt, E. Jeansson, M. Ishii, F. F. Pérez, T. Suzuki & S. Watelet, 2016. A new global interior ocean mapped climatology: the 1°x1° GLODAP version 2. Earth Syst. Sci. Data, 8, 325-340, doi:10.5194/essd-8-325-2016.

Lo Monaco C., C. Goyet, N. Metzl, A. Poisson and F. Touratier, 2005. Distribution and inventory of anthropogenic CO2 in the Southern Ocean : comparison of three data-based methods. Journal Geophys. Res. . 110, C09S02, doi:10.1029/2004JC002571.

Mahieu, L., Lo Monaco, C., Metzl, N., Fin, J., and Mignon, C.: Variability and stability of anthropogenic CO2 in Antarctic Bottom Water observed in the Indian sector of the Southern Ocean, 1978–2018, Ocean Sci., 16, 1559–1576, https://doi.org/10.5194/os-16-1559-2020, 2020.

Metzl, N., et al: Anthropogenic CO2, air–sea CO2 fluxes, and acidification in the Southern Ocean: results from a time-series analysis at station OISO-KERFIX (51° S–68° E), Ocean Sci., 20, 725–758, https://doi.org/10.5194/os-20-725-2024, 2024.

Müller, J. D. , N. Gruber, B. Carter, R. Feely, M. Ishii, N. Lange, S. K. Lauvset, A. Murata, A. Olsen, F. F. Pérez, C. Sabine, T. Tanhua, R. Wanninkhof, D. Zhu, Decadal trends in the oceanic storage of anthropogenic carbon from 1994 to 2014. AGU Adv. 4, e2023AV000875 (2023).

Orsi, A. H., T. Whitworth III, and W. D. Nowlin Jr., On the meridional extent and fronts of the Antarctic Circumpolar Current, Deep Sea Res., Part I, 42, 641-673, 1995.

Pardo, P. C., et al,: Carbon uptake and biogeochemical change in the Southern Ocean, south of Tasmania. Biogeosciences, 14(22), 5217–5237. https://doi.org/10.5194/bg-14-5217-2017, 2017

Ríos, A. F., Velo, A., Pardo, P. C., Hoppema, M., and Pérez, F. F.: An update of anthropogenic CO2 storage rates in the western South Atlantic basin and the role of Antarctic Bottom Water, J. Mar. Syst., 94, 197–203, https://doi.org/10.1016/j.jmarsys.2011.11.023, 2012.

Rödenbeck, C., et al, 2015. Data-based estimates of the ocean carbon sink variability – First results of the Surface Ocean pCO2 Mapping intercomparison (SOCOM). Biogeosciences 12: 7251-7278. doi:10.5194/bg-12-7251-2015..

Vazquez-Rodriguez, M., et al, 2009. Anthropogenic Carbon Distributions in the Atlantic Ocean: databased estimates from the Arctic to the Antarctic. Biogeosciences, 6, 439-451. https://doi.org/10.5194/bg-6-439-2009

We sincerely appreciate your careful and constructive review, as well as the inclusion of relevant references that greatly assisted us in refining and improving the manuscript. We carefully reviewed each cited paper and, after evaluating their relevance to our study, made the following adjustments to the reference list and corresponding text:

(1) Replaced *Zemskova et al., 2022* with **Rios et al., 2012**, **Pardo et al., 2017**, and **Mahieu et al., 2020** (Line 46).

(2) Added **Lauvset et al, 2016** (Line 68).

(3) Replaced Hauck et al., 2023 with **Hauck et al., 2023** and **Lo Monaco et al., 2005** (Line 71).

(4) Added the previously missing references **Zhang et al., 2023** and **Metzl et al., 2024** (Line 104).

(5) Added **Gruber et al., 1998** (Line 197).

(6) For citations of the TrOCA method in the Southern Ocean, included **Metzl et al., 2024**; **Vázquez-Rodríguez et al., 2009**; and **Mahieu et al., 2020** (Line 205).

(7) Updated the CO2SYS reference to **Sharp et al., 2023** (Line 214).

(8) Added **Orsi et al., 1995** in the caption of Figure 7, 8, A4, A5.

**Response to Referee #2 (R2):**

Zhong et al. developed a new ocean carbonate data product for the Southern Ocean by integrating float- and ship-based observations. Hydrographic data from Argo floats were used to approximate carbonate variables using established algorithms, which were then merged with ship-based measurements. As the Southern Ocean remains one of the least sampled regions for carbonate system variables, this product represents an important step toward filling that gap. It is a valuable contribution, and the paper is well written.

Thank you for your positive comments, we are pleased that the main messages and importance of the study come across. The comments below have been extremely helpful to refine our work. The line numbers mentioned in our responses refer to the revised version with *Track Changes* enabled for review.

1. Title and Abstract
The title and abstract are misleading in their current form. The float data include temperature, salinity, and in some cases oxygen, which are then used to reconstruct ocean carbon variables. As written, readers could be misled to believe that the float data directly provide ocean carbonate variables, which is not accurate. Please revise the wording to more clearly reflect the indirect nature of the reconstruction.

Thank you for raising this important point. To address this point, and consistent with Reviewer 1's suggestion, we have revised the title to:

"*Climatological fields of Southern Ocean interior carbonate system parameters and anthropogenic $CO_2$ reconstructed and integrated from float- and ship-based observations*".

2. Gridding method
The authors mentioned that "Profile data for each parameter are sorted into spatial bins of 1° latitude × 1° longitude bins and 84 vertical pressure levels to generate homogenized three-dimensional gridded products." As we all know, there aren't enough data at each of the grid points for the global ocean. Did the authors use some kind of gridding method? More details are needed.

Thank you for this helpful comment, and we apologize for the lack of clarity in Section 3.3. To clarify, our gridding procedure follows a straightforward bin-averaging approach without any horizontal interpolation or objective mapping. Specifically, (1) each Argo float profile is vertically interpolated to the predefined 84 depth levels; (2) extreme outliers are identified and removed; and (3) all valid measurements within each grid cell and depth level are averaged to obtain representative values. Grid cells without valid data are left empty. We have added a clarifying sentence at the end of Section 3.3, which now reads:

"*Finally, for each grid cell, all valid measurements are averaged to obtain representative values, while cells without observations are left empty. This bin-*

*averaging approach ensures that the gridded products are entirely observation-based and preserve the genuine spatial structure of the compiled dataset."* (Line 244)

3. Spatial Grid Convention

It is more standard to use the grid ordering of longitude, latitude, and depth, rather than the current latitude, longitude, and depth. Adopting the conventional structure will facilitate easier integration with other datasets.

We have adopted the recommended structure.

4. Vertical Resolution

(a) Use depth in meters rather than pressure in dbars, in line with most other ocean carbon data products.

(b) Adopt standardized depth levels consistent with the World Ocean Atlas (WOA). For reference, the current recommended levels can be either 33 (not presented here) or 102 (as below):

0, 5, 10, 15, 20, 25, 30, 35, 40, 45, 50, 55, 60, 65, 70, 75, 80, 85, 90, 95, 100, 125, 150, 175, 200, 225, 250, 275, 300, 325, 350, 375, 400, 425, 450, 475, 500, 550, 600, 650, 700, 750, 800, 850, 900, 950, 1000, 1050, 1100, 1150, 1200, 1250, 1300, 1350, 1400, 1450, 1500, 1550, 1600, 1650, 1700, 1750, 1800, 1850, 1900, 1950, 2000, 2100, 2200, 2300, 2400, 2500, 2600, 2700, 2800, 2900, 3000, 3100, 3200, 3300, 3400, 3500, 3600, 3700, 3800, 3900, 4000, 4100, 4200, 4300, 4400, 4500, 4600, 4700, 4800, 4900, 5000, 5100, 5200, 5300, 5400, 5500 m.

(1) We have now revised the dataset to use depth in meters rather than pressure.

(2) Following your suggestion, we have adopted a standardized depth structure consistent with the World Ocean Atlas (WOA). However, since Argo floats have a coarser vertical sampling interval than ship-based measurements below 2000 m, directly applying the WOA 100 m interval beyond this depth would result in many missing values. To balance vertical resolution with data completeness, we therefore defined 84 depth levels as follows: 0–100 m at 5m intervals (20 levels), 100–500 m at 25m intervals (15 levels), 500–2000 m at 50m intervals (30 levels, consistent with the WOA above), and 2000–5600 m at 200 m intervals (19 levels). We revise manuscript and Figure 3a has been updated accordingly to reflect this revised depth scheme:

[Figure]

**5. Product Variants**

Presenting multiple output variants may confuse end-users. I recommend presenting the final merged grid product as the primary output, supplemented by quality flags that indicate the provenance of each grid point (e.g., derived from non-O2 floats, O2 floats, shipboard observations, etc.). This approach maintains transparency while simplifying use.

Yes, that is a good suggestion. We have adopted the *All-Data Grid* as the primary product to simplify data usage while maintaining transparency. For each grid cell, we additionally provide the corresponding values derived from non-$O_2$ floats, $O_2$ floats, and shipboard observations, allowing users to trace the provenance of each data point as needed.

**6. Uncertainty Estimates**

The reported uncertainties appear unrealistically low. This is concerning, as the ESPER algorithm alone typically introduces uncertainties on the order of ~20 μmol/kg for DIC and TA, not including additional uncertainties from spatial gridding. Please revisit your uncertainty estimation procedure.

We appreciate the opportunity to clarify this point. In the ESPER_NN framework, uncertainties are estimated using one standard deviation and interpolated in depth–salinity space based on baseline error estimates. When validated against GLODAPv2.2020 measurements, the reported global RMSE ranges from 4.5 to 13.2 μmol kg$^{-1}$, 3.7 to 5.2 μmol kg$^{-1}$ for TA, and 4.8 to 16.7 μmol kg$^{-1}$ for DIC (Carter et al., 2021).

In this study, we further assessed ESPER_NN uncertainties using GLODAPv2.2023 dataset, which includes a larger number of independent measurements not used in the training group. As shown in Table 2, the all-water-column averaged uncertainties in the Southern Ocean from ESPER_NN are 4.1 μmol kg$^{-1}$ for TA (5.21 μmol kg$^{-1}$

without oxygen) and 6.61 µmol kg⁻¹ for DIC (9.11 µmol kg⁻¹ without oxygen). These uncertainties are not uniform with depth; they generally decrease with depth and are largest in the surface layer where biological processes and air–sea exchange are strong.

Although uncertainties in Argo-measured temperature, salinity, and oxygen are generally higher than those from shipboard observations—particularly for oxygen (assumed 2 %)—the reconstructed TA and DIC remain relatively robust. In our quality control procedure, we combined the Argo measurement QC flags with the 2σ uncertainties derived from the assessment evaluation for all parameter estimates. Across all Argo floats used, the maximum estimated uncertainties are 8.2 µmol kg⁻¹ for TA (Figure R1a) and 17.2 µmol kg⁻¹ for DIC (Figure R1b).

The accompanying profile plots show that uncertainties below ~300 m are relatively small. Consequently, the uncertainty propagated to $C_{ant}$ through the Monte Carlo approach remains moderate (shown in blue in Figure R1). It is estimated to be about ±4–6 µmol kg⁻¹ in both the full-parameter and hydrography-only pathways (Section 4.5, Line 463). This value represents the averaged uncertainty calculated over the full water column. Finally, in the gridded product, we also quantify additional uncertainty sources associated with spatial gridding and include the standard deviation of each variable in the output fields.

We have added the following explanatory sentence in the main text:

"*Based on the estimated RMSE of ESPER_NN, float-derived estimates are expected to fall within twice the model's estimated uncertainty range, serving as a criterion for the quality control applied to the Argo float dataset.*" (Line 336)

[Figure]

**Figure R1**. Vertical distribution of estimated uncertainties for total alkalinity (TA), dissolved inorganic carbon (DIC), and anthropogenic carbon ($C_{ant}$) along two representative Argo float profiles.

**7. Data Publication and Metadata**

The product is currently published on Mendeley, but with very limited metadata. For long-term archiving, discoverability, and broader community uptake, I strongly encourage the authors to submit the dataset to NOAA's Ocean Carbon and

Acidification Data System (OCADS) at NCEI. This would ensure proper long-term archiving, rich metadata documentation, and alignment with established practices in the ocean carbon community. Most importantly, it will make this data product available along with similar ocean carbonate data products.

Yes, we agree with this recommendation. The dataset has been submitted to NOAA's Ocean Carbon and Acidification Data System (OCADS) at NCEI to ensure long-term archiving, comprehensive metadata documentation, and consistency with community standards. The submission is currently under review by OCADS.

[Figure]

**8. Output Variables**

At present, the output only includes TA, DIC, pH, anthropogenic carbon, and aragonite saturation state. I recommend also reporting additional ocean carbonate system variables, such as the fugacity of carbon dioxide, carbonate ion concentration, calcite saturation state, and the Revelle Factor. These are commonly provided in comparable products and would broaden the product's utility.

Thank you for this helpful suggestion. Based on the variables already included in our dataset, we simultaneously derived additional carbonate system parameters—namely, carbonate ion concentration, calcite saturation state, and the Revelle Factor—using the CO2SYS program, and these variables have now been added to the product. Regarding the fugacity of carbon dioxide ($fCO_2$), this study primarily targets the ocean interior. Since $fCO_2$ (or equivalently $pCO_2$) is mainly applied at the air–sea interface for estimating $CO_2$ fluxes, and considering that the ESPER model is designed for reconstructing seawater biogeochemical properties rather than $pCO_2$ fields, we decided not to include $fCO_2$ in this dataset after careful consideration and discussion. We believe this decision maintains the focus of the dataset while ensuring broad usability.

**Minor Comments**

Throughout the manuscript, please use the term "variables" instead of "parameters", to describe an observed property.

We thank the reviewer for this helpful suggestion. Following your advice, we have replaced "parameters" with "variables" throughout the manuscript when describing observed properties. The only exception is the commonly used phrase "carbonate system parameters", which we have kept unchanged given its frequent usage in the field.